# A viral effector blocks the turnover of a plant NLR receptor to trigger a robust immune response

Chunli Wang[1,3], Min Zhu[1,3], Hao Hong [ID][1,3], Jia Li [ID][1,3], Chongkun Zuo [ID][1], Yu Zhang[1], Yajie Shi [ID][1], Suyu Liu[1], Haohua Yu[1], Yuling Yan[1], Jing Chen [ID][1], Lingna Shangguan[1], Aiping Zhi[1], Rongzhen Chen[1], Karen Thulasi Devendrakumar [ID][2] & Xiaorong Tao [ID][1✉]

## Abstract

Plant intracellular nucleotide-binding and leucine-rich repeat immune receptors (NLRs) play a key role in activating a strong pathogen defense response. Plant NLR proteins are tightly regulated and accumulate at very low levels in the absence of pathogen effectors. However, little is known about how this low level of NLR proteins is able to induce robust immune responses upon recognition of pathogen effectors. Here, we report that, in the absence of effector, the inactive form of the tomato NLR Sw-5b is targeted for ubiquitination by the E3 ligase SBP1. Interaction of SBP1 with Sw-5b via only its N-terminal domain leads to slow turnover. In contrast, in its auto-active state, Sw-5b is rapidly turned over as SBP1 is upregulated and interacts with both its N-terminal and NB-LRR domains. During infection with the tomato spotted wilt virus, the viral effector NSm interacts with Sw-5b and disrupts the interaction of Sw-5b with SBP1, thereby stabilizing the active Sw-5b and allowing it to induce a robust immune response.

**Keywords** Pathogen Effector; NLR Receptor; Effector-triggered Immunity; E3 Ligase; Homeostasis
**Subject Categories** Microbiology, Virology & Host Pathogen Interaction; Plant Biology

## Introduction

The plant immune system is important for plant to ward off pathogens and to survive in natural ecosystems (Jones and Dangl, 2006; Boller and He, 2009; Jones et al, 2016; Couto and Zipfel, 2016; Zhou and Zhang, 2020). Plants utilize membrane localized pattern recognition receptors (PRRs) to detect conserved microbe/pathogen-associated molecular patterns (MAMPs/PAMPs) and activate PAMP-triggered immunity (PTI). Plants also utilize another class of intracellular receptors called nucleotide-binding (NB) and leucine-rich repeat (LRR) receptors (NLRs) to detect pathogen effectors and activates a strong immune response known as effector triggered immunity (ETI) (Dodds and Rathjen, 2010; Cui et al, 2015; Li et al, 2015; Cesari, 2018; Kourelis and van der Hoorn, 2018). ETI is often accompanied by hypersensitive response (HR), a localized programmed cell death (PCD), at the infection sites (Cui et al, 2015; Saur et al, 2020). NLRs are the major group of resistance genes in plants and are among the largest and the most diversified gene families in plant genomes (Li et al, 2015; Kapos et al, 2019; Jones et al, 2016). NLRs provide resistance to a variety of pathogens. However, their abundance in plant genomes and their capacity to induce strong immune responses necessitates precise and timely control of NLR accumulation and activation. Such stringent controls are necessary to avoid massive fitness costs due to NLR activation in the absence of pathogens and to ensure appropriate immune responses upon pathogen attack (Tian et al, 2003; Deng et al, 2017).

Ubiquitination is an important post-translational modification for the regulation of protein stability in Eukaryotes (Smalle and Vierstra, 2004; Vierstra, 2009; Zhou and Zeng, 2017). Ubiquitination is mediated by three major enzymes: the E1 ubiquitin-activating enzyme, E2 ubiquitin-conjugating enzyme, and E3 ubiquitin ligase. They act together to add ubiquitin (Ub) to specific substrate proteins (Pickart, 2001). These ubiquitinated proteins are then targeted for degradation by the 26S proteasome. E3 ligases are the major determinants of substrate specificity during ubiquitination. The Arabidopsis genome encodes more than 1400 E3 ligases to regulate the homeostasis of proteins involved in diverse pathways (Smalle and Vierstra, 2004; Vierstra, 2009) including proteins involved in plant immunity (Chen and Li, 2012). Arabidopsis U-box E3 ligases PUB12 and PUB13 regulate the protein levels of the PRR FLS2, consequently modulating FLS2-mediated immunity (Lu et al, 2011). The homeostasis of the immune kinase BIK1 was found to be dynamically controlled by a regulatory E3 module (Monaghan et al, 2014; Wang et al, 2018). CPR1, an F-box containing E3 ligase, was shown to regulate the protein turnover of Arabidopsis NLR SNC1 (Cheng et al, 2011). Although the full spectrum of CPR1 substrates is still unknown, CPR1 was also shown to regulate the protein stability of Arabidopsis NLR RPS2 (Huang et al, 2016). A RING-type E3 ligase MIR1 was found to regulate the stability of MLA, an NLR

[1]The Key Laboratory of Plant Immunity, Department of Plant Pathology, Nanjing Agricultural University, Nanjing 210095, P. R. China. [2]Department of Botany and Michael Smith Laboratories, University of British Columbia, Vancouver, BC V6T 1Z4, Canada. [3]These authors contributed equally: Chunli Wang, Min Zhu, Hao Hong, Jia Li. ✉E-mail: taoxiaorong@njau.edu.cn

protein, and MLA-mediated immunity to powdery mildew (Wang et al, 2016). Strikingly E3 ligases SNIPER1 and SNIPER2 broadly regulate the homeostasis of diverse NLR immune receptors, including SNC1, SUMM2, RPP4, and CHS1 in *Arabidopsis thaliana* (Wu et al, 2020). A TurboID-based proximity labeling method identified a U-box type E3 ubiquitin ligase UBR7 that interacts with tobacco NLR N. Overexpression of UBR7 reduced the stability of N, whereas silencing of UBR7 enhanced the *N*-mediated resistance to tobacco mosaic virus (TMV) (Zhang et al, 2019).

NLR immune receptors function as molecular switches of plant defense (Takken et al, 2006; Lukasik and Takken, 2009; Williams et al, 2011; Wang et al, 2019; Ma et al, 2020). In the absence of pathogen effectors, NLRs are maintained in an inactive, ADP-bound state. Upon recognition of pathogen effectors, NLRs switch to an active, ATP-bound state. It has been reported that NLR ADP-bound and ATP-bound states are in an equilibrium (Bernoux et al, 2016). This equilibrium allows for a small number of self-active NLR receptors in the absence of pathogen effectors. Recognition of pathogen effectors causes the equilibrium to be shifted from the ADP-bound, inactive to the ATP-bound, active state, thereby activating immune responses. How the homeostasis between the ON and OFF state of NLRs is maintained in the presence or absence of their cognate effector is still poorly understood. NLRs are found in very low levels in steady state conditions, likely to avoid fitness costs in the absence of pathogens. It remains obscure how recognition of pathogen effectors converts low level of NLRs to trigger robust immune responses.

The tomato spotted wilt orthotospovirus (TSWV) is a highly destructive plant viral pathogen in the world (Scholthof et al, 2011). It infects more than 1000 species of plants including various commercial important crops such as tomatoes, peppers, peanuts, lettuce, and flowers (Oliver and Whitfield, 2016). In tomato, Sw-5b is the most effective resistance gene used to control TSWV (Spassova et al, 2001; Turina, 2016; Zhu et al, 2019). Sw-5 was derived from *Solanum peruvianum* in South America and was widely introduced into different tomato cultivars. Sw-5 locus has five paralogs, Sw-5a to Sw-5e, on the long arm of chromosome 9 (Brommonschenkel et al, 2000; Spassova et al, 2001; Turina, 2016). Of these, only Sw-5b was shown to confer effective resistance against TSWV (Spassova et al, 2001). Sw-5b encodes an NLR protein with coil-coil (CC), NB, and LRR domains (Brommonschenkel et al, 2000; Spassova et al, 2001). In addition, Sw-5b and its close homologs R8, Mi-1.2, Hero, and Rpi-blb2 also contain a *Solanaceae* domain (SD) at their N-termini (Lukasik-Shreepaathy et al, 2012; Vossen et al, 2016; van der Vossen et al, 2005). Sw-5b recognizes NSm, a movement protein encoded by TSWV, and triggers ETI (Hallwass et al, 2014; Peiró et al, 2014; Zhao et al, 2016). A conserved 21-amino acid peptide region within the NSm (NSm[21]) is sufficient to trigger Sw-5b-mediated HR (Zhu et al, 2017). In the absence of NSm or NSm[21], Sw-5b has multilayered regulation to maintain it in an inactive state in which Sw-5b LRR domain suppresses the central NB-ARC, and the CC and SD domains further additively suppress the NB-ARC-LRR (Chen et al, 2016). Both the LRR and SD domains of Sw-5b are required for NSm recognition (Li et al, 2019). In the presence of NSm or NSm[21], Sw-5b SD domain recognizes NSm and relieve the inhibitory effects of the SD and CC on NB-ARC-LRR. The LRR domain then recognizes NSm and depresses the suppression of LRR on the NB-ARC, leading to the switch activation of Sw-5b from an inactive

state to an active state (Chen et al, 2016; Zhu et al, 2017; Li et al, 2019).

In this study, using the Sw-5b NLR and the tospoviral NSm effector as the model system, we demonstrate that E3-mediated turnover of inactive/active state of Sw-5b NLR is blocked by the viral pathogen effector to induce a robust immunity. Protein stability of Sw-5b is regulated by the 26S proteasome degradation pathway and the viral effector NSm can protect Sw-5b from 26S proteasome-mediated protein turnover. Using a yeast two-hybrid screen, we identified SBP1, a RING-type E3 ligase as a direct interactor of both the SD and NB-LRR domains of Sw-5b. SBP1 is capable of ubiquitinating Sw-5b and knockout of *SBP1* enhances the accumulation of Sw-5b, and its immunity to TSWV. Strikingly, the active form of Sw-5b is degraded much faster than its inactive form. We show that the NSm effector that is recognized by both Sw-5b SD and NB-LRR domains disrupts the interaction of both the aforementioned Sw-5b regions with SBP1 leading to the accumulation of active form of Sw-5b and robust resistance. Based on these findings, we propose that in the absence of NSm effector, active-state Sw-5b NLR is rapidly ubiquitinated and degraded via the 26S proteasome-dependent pathway. However, once Sw-5b recognizes and binds the NSm effector, the active form of NLR is stabilized and accumulates to levels sufficient to elicit a robust immune response.

# Results

## TSWV effector NSm protects Sw-5b NLR from 26S proteasome-mediated degradation

When analyzing the stability of Sw-5b in the presence or absence of the TSWV effector NSm, we observed that the protein accumulation of Sw-5b could be significantly enhanced in the presence of viral NSm prior to the onset of cell death (20 to 28 h post infiltration, hpi). To validate this observation, we co-expressed YFP-Sw-5b with pCambia2300S empty vector (EV) in one-half of a *Nicotiana benthamiana* leaf and co-expressed YFP-Sw-5b with TSWV NSm in the other half of the leaf and examined the YFP-Sw-5b levels by immunoblot. The results showed that the YFP-Sw-5b protein levels were significantly increased in the presence of TSWV effector NSm without or with cycloheximide (CHX) treatment (Fig. 1A; Appendix Fig. S1A). Next, we used a tomato zonate spot orthotospovirus (TZSV) encoded NSm variant that is not recognized by Sw-5b in the assay (Zhu et al, 2017). As shown in Fig. EV1A, the YFP-Sw-5b protein levels were not elevated by non-elicitor TZSV NSm. To examine whether the low Sw-5b protein level in the absence of TSWV NSm is due to its turnover through the 26S proteasome degradation pathway, we expressed YFP-Sw-5b in the presence or absence of the proteasome inhibitor MG132 in the two halves of the same *N. benthamiana* leaf. The immunoblot results showed that the protein accumulation of YFP-Sw-5b was significantly enhanced in the presence of MG132 (Fig. 1B), suggesting that Sw-5b is degraded through the 26S proteasome pathway. YFP-Sw-5b protein level in the presence of TSWV NSm is comparable to the YFP-Sw-5b protein level with addition of MG132 (Appendix Fig. S1B). A Sw-5b homolog from TSWV susceptible tomato cultivar Heinz1706 (henceforth referred to as susceptible/non-resistant Sw-5b[Heinz]) does not recognize TSWV

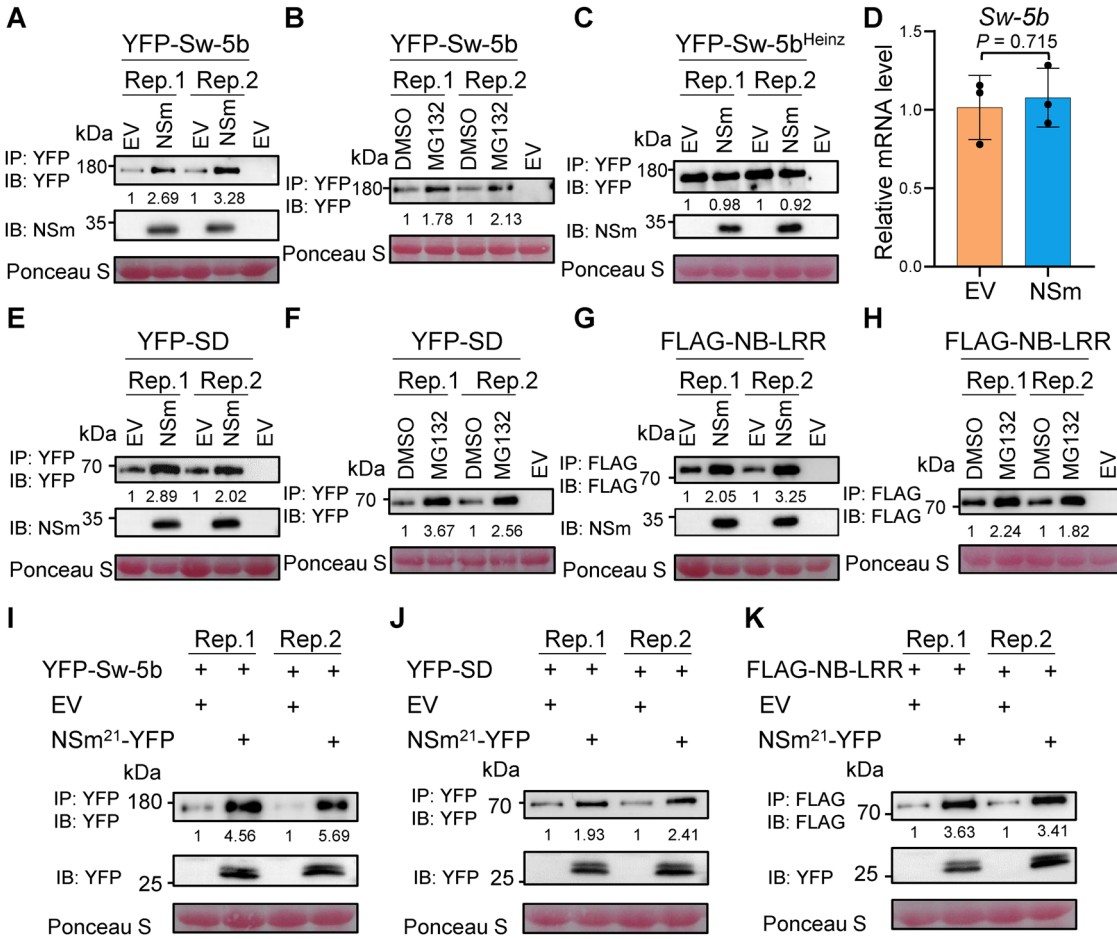

**Figure 1. Tospoviral NSm protects Sw-5b NLR from plant 26S proteasome-mediated protein turnover.**

(A, C, E, G) Protein accumulation of Sw-5b, Sw-5b^Heinz homolog, SD, and NB-LRR without or with NSm in *N. benthamiana* leaves at 21 h post infiltration (hpi). YFP-Sw-5b (**A**), YFP-Sw-5b^Heinz (**C**), YFP-SD (**E**), or FLAG-NB-LRR (**G**) was co-expressed with the p2300 empty vector (EV) in one-half leaf of *N. benthamiana* plant and co-expressed with NSm in another half of the same leaf. Samples were collected for immunoblotting at 21 hpi. (**B, F, H**) Protein accumulation of Sw-5b, SD, and NB-LRR without or with MG132 in *N. benthamiana* leaves at 21 hpi. YFP-Sw-5b (**B**), YFP-SD (**F**), or FLAG-NB-LRR (**H**) was expressed without MG132 in one-half of a *N. benthamiana* leaf and expressed with 25 μM MG132 in another half of the same leaf. DMSO or MG132 was infiltrated at 13 hpi and samples were collected for immunoblotting at 21 hpi. (**D**) Relative RNA expression level of YFP-Sw-5b without or with NSm in *N. benthamiana* leaves. YFP-Sw-5b was co-expressed with EV or NSm and their RNA expression was analyzed by qRT-PCR at 21 hpi. Data are presented as means ± SD (*n* = 3 biologically independent samples). Data were analyzed by two-sided Student's *t*-test. (**I–K**) Protein accumulation of Sw-5b, SD, and NB-LRR without or with NSm^21 peptide in *N. benthamiana* leaves at 21 hpi. YFP-Sw-5b (**I**), YFP-SD (**J**), or FLAG-NB-LRR (**K**) was co-expressed with the EV in one-half leaf of *N. benthamiana* plant and co-expressed with NSm^21-YFP in the other half of the same leaf. Samples were collected for immunoblotting at 21 hpi. The blots were detected using YFP, FLAG, and NSm-specific antibodies. Rep. 1 and Rep. 2 refer to two independent repeated results of the experiments. Protein accumulation level was quantified by ImageJ software. IB, immunoblot with specific antibody; IP, immunoprecipitation with specific antibody. Ponceau S staining was used to show the amount of protein loaded. NSm refers to the elicitor TSWV NSm. Source data are provided as a Source data file. All experiments were repeated at least three times with similar results. Source data are available online for this figure.

NSm (Zhu et al, 2017; Li et al, 2019). To examine whether the protein stability of Sw-5b^Heinz is affected by the presence of TSWV NSm, YFP-Sw-5b^Heinz was co-expressed with EV in one-half of a *N. benthamiana* leaf and co-expressed with TSWV NSm in the other half of the same leaf. Immunoblot analysis showed that the protein accumulation of YFP-Sw-5b^Heinz was not enhanced in the presence of TSWV NSm (Fig. 1C). In addition, the transcript level of *Sw-5b* or *Sw-5b^Heinz* was not affected by TSWV NSm (Fig. 1D; Appendix Fig. S1C). These data suggest that the protein stability of Sw-5b NLR is regulated by the 26S proteasome degradation pathway and TSWV effector NSm can protect Sw-5b from 26S proteasome-mediated protein turnover.

We recently showed that both the SD and NB-LRR domains of Sw-5b play an important role in the recognition of TSWV NSm (Zhu et al, 2017; Li et al, 2019). Similar to the full-length Sw-5b, the accumulation of YFP-SD and FLAG-NB-LRR was significantly enhanced in the presence of elicitor TSWV NSm, but not non-elicitor TZSV NSm (Figs. 1E,G and EV1B,C; Appendix Fig. S1D,G). In addition, the protein levels of both YFP-SD and FLAG-NB-LRR were significantly increased in the presence of MG132 (Fig. 1F,H), suggesting that both SD and NB-LRR proteins are targeted to the 26S proteasome for degradation in the absence of viral effector. In addition, the protein level of SD and NB-LRR in the presence of TSWV NSm is comparable to their level with addition of MG132

(Appendix Fig. S1E,H). The transcript level of *Sw-5b SD* and *NB-LRR* was not affected by TSWV NSm (Appendix Fig. S1F,I). Taken together, these data suggest that TSWV NSm protects both SD and NB-LRR from 26S proteasome-mediated degradation.

Our previous studies showed that a minimal 21-amino acid peptide region of TSWV NSm (NSm[21]) is sufficient for Sw-5b to trigger HR (Li et al, 2019; Zhu et al, 2017). To examine whether NSm[21] can protect full-length Sw-5b or its SD and NB-LRR domains from degradation, we expressed them with or without NSm[21]-YFP in *N. benthamiana* leaves. Immunoblot results showed that NSm[21]-YFP expression increased protein accumulation of Sw-5b, and its SD and NB-LRR domains without altering their transcript level (Fig. 1I–K; Appendix Fig. S1J–L).

## An E3 ligase SBP1 interacts with both SD and NB domains of Sw-5b NLR

Since the proteasome is involved in the homeostasis of Sw-5b, an E3 ligase is likely targeting Sw-5b for ubiquitination and subsequent degradation. To identify candidate E3 ligase(s) responsible for regulating the homeostasis of Sw-5b, we performed a yeast two-hybrid (Y2H) screen against a *N. benthamiana* cDNA library using Sw-5b NB domain as bait. One of the candidates was the S-ribonuclease binding protein 1 (SBP1, Niben101Scf00868Ctg019) which encodes a putative RING-type E3 ubiquitin ligase (Appendix Fig. S2A,B) and it was repeatedly identified in the screen. We also identified SBP1 E3 ligase as an interacting protein in an independent Y2H screen using the Sw-5b SD domain as bait (Appendix Fig. S2A). The full-length SBP1 contains 337 amino acids while the clone identified from the Y2H screen contained amino acids 1–306 and lacked the 31 C-terminal amino acids. The full-length *SBP1* gene was cloned from the cDNA of *N. benthamiana*. A targeted Y2H assay confirmed that SBP1 directly interacts with both the SD and NB domains of Sw-5b (Fig. 2A).

To confirm the interaction of SBP1 with the SD and NB domains of Sw-5b, we further performed split-luciferase complementation (SLC) and co-immunoprecipitation (Co-IP) assays. The SLC assay showed that SBP1-nLUC interacts with cLUC-SD or with cLUC-NB to produce luciferase activity in *N. benthamiana* leaves suggesting that SBP1 does indeed interact with SD and NB domains of Sw-5b (Fig. 2B). Considering that overexpression of the E3 ligase SBP1 may hasten the degradation of the Sw-5b SD and NB domains, we generated a loss-of-function (LOF) mutant of SBP1, SBP1[RM] (SBP1 RING domain mutant) in which the conserved Cys-306 and Cys-312 amino acid residues of the RING domain were both mutated to Ser (Appendix Fig. S2B) (Garcia-Barcena et al, 2020). These mutations abolish the ubiquitin ligation activity of the RING E3 but still allow it to interact with its substrate protein. Indeed, co-expression of cLUC-SD or cLUC-NB with SBP1[RM]-nLUC generated a stronger luciferase activity compared to their co-expression with wild-type SBP1-nLUC (Fig. 2B; Appendix Fig. S2C). In targeted Co-IP assays, both SD-FLAG and FLAG-NB-LRR co-immunoprecipitate with SBP1[RM]-RFP but not with RFP alone (Fig. 2C,D). Further, full-length YFP-Sw-5b but not the non-resistant YFP-Sw-5b[Heinz] homolog co-immunoprecipitated with SBP1[RM]-RFP (Fig. 2E), suggesting that SBP1 specifically interacts with the resistant Sw-5b. Taken together, SBP1 directly interacts with Sw-5b via its SD and NB domains while it is incapable of interacting with the non-resistant Sw-5b[Heinz].

## SBP1 regulates Sw-5b stability and Sw-5b-mediated immunity

There are two close homologs of *SBP1*, *SBP1-1* (Niben101Scf00868Ctg019) and *SBP1-2* (Niben101Scf04995Ctg015), in the genome sequence of *N. benthamiana* plant. *SBP1-1* is the gene that was identified from Y2H screen. *SBP1-1* and *SBP1-2* have 97.44% nucleotide sequence identity. To further investigate the role of the putative E3 ligase, SBP1, in the homeostasis of Sw-5b protein levels, we generated two knock out (KO) *N. benthamiana* mutant lines *sbp1-1* and *sbp1-2* in which both *SBP1-1* and *SBP1-2* are knocked out (Appendix Fig. S3A,B). We transiently expressed YFP-Sw-5b in wild-type (WT), *sbp1-1,* and *sbp1-2* mutant plant leaves and the protein accumulation were analyzed by immunoblots. The results showed that the protein level of YFP-Sw-5b is significantly higher in both *sbp1-1* and *sbp1-2* leaves compared to WT plant leaves (Figs. 3A and EV1D). However, the protein level of YFP-Sw-5b[Heinz] was not increased in *sbp1-1* or *sbp1-2* leaves compared to WT leaves (Figs. 3B and EV1E). Further, Real-time quantitative reverse transcription polymerase chain reaction (Real-time qRT-PCR) results confirmed that the YFP-Sw-5b and YFP-Sw-5b[Heinz] were expressed to similar levels in WT, *sbp1-1* and *sbp1-2* and so the differences seen in the protein level are not due to transcriptional differences (Appendix Fig. S4A,B). We also used a similar strategy to examine whether the SD or NB-LRR protein accumulated more in *SBP1*-KO plant leaves compared to WT leaves. The results showed that protein level of both YFP-SD and FLAG-NB-LRR was increased in *sbp1-1* and *sbp1-2* in comparison to their protein level in WT (Figs. 3C,D and EV1F,G). The *SBP1*-KO did not alter the transcript level of SD and NB-LRR (Appendix Fig. S4C,D).

We next examined the effect of *SlSBP1* in *Solanum lycopersicum* on the accumulation of Sw-5b. *Sl*SBP1 shared 87.54% amino acid sequence identity with *Nb*SBP1 from *N. benthamiana* (Appendix Fig. S5A). We knocked down (KD) the expression of *SlSBP1* in tomato plants carrying the *Sw-5b* resistance gene using TRV induced gene silencing (*SlSBP1*-KD) (Appendix Fig. S5B–D). Immunoblot assay failed to detect endogenous Sw-5b protein in tomato using Sw-5b specific antibodies. This is probably because the endogenous Sw-5b protein level is too low to be detectable. Therefore, the total protein from TRV-*GUS* and TRV-*SlSBP1* tomato plants were precipitated with Trichoroacetic Acid (TCA)-Acetone and the enriched total proteins were analyzed by immunoblot using specific antibodies against Sw-5b CC-NB domains. Repeated assays showed that, compared to the control tomato plants, the protein levels of Sw-5b was significantly increased in *SlSBP1*-knockdown plants (Fig. 3E), suggesting *Sl*SBP1 does regulate the accumulation of Sw-5b.

Further, we tested the substrate spectrum of SBP1. R8, Mi-1.2, Rpi-blb2, and Hero are solanaceae CC-NLRs that contain an extra SD domain similar to Sw-5b. Of these, R8 and Rpi-blb2 were further used to test the substrate spectrum of SBP1. Sw-5b has 83.1% and 30% amino acid sequence similarity, respectively, with R8 and Rpi-blb2. To investigate whether SBP1 plays a role in regulating the homeostasis of R8 and Rpi-blb2, we transiently expressed YFP-R8 and FLAG-Rpi-blb2 in WT, *sbp1-1* and *sbp1-2* *N. benthamiana* leaves. From immunoblot assays it was seen that the protein level of YFP-R8 but not FLAG-Rpi-blb2 was increased in *sbp1-1* and *sbp1-2* leaves compared to their protein level in WT leaves (Fig. 3F,G), and *SBP1*-KO has no effect on YFP-R8 or FLAG-

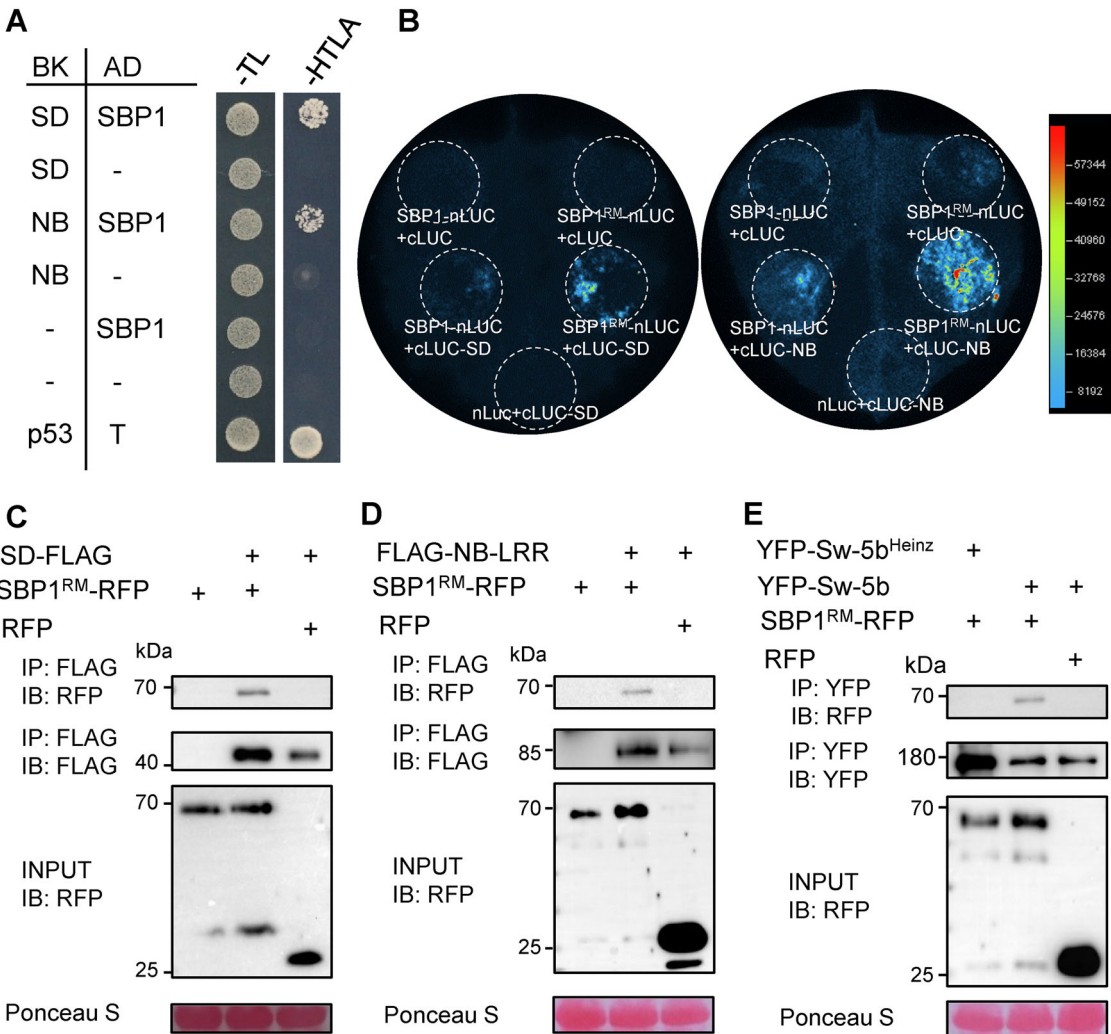

**Figure 2. E3 ligase SBP1 is an interactor of full-length Sw-5b, and its SD and NB domains.**

(A) The interaction of E3 ligase SBP1 with SD and NB domains of Sw-5b analyzed by yeast two-hybrid (Y2H) assay. AD-SBP1 or AD empty vector (pGADT7) was used to assay for the interaction with BK-NB, BK-SD or BK empty vector (pGBKT7). AD-T (pGADT7-T) and BK-p53 (pGBKT7-p53) were used as positive control. (B) Interaction of Sw-5b NB and SD with SBP1 analyzed by split-luciferase complementation assay in *N. benthamiana* leaves. cLUC-SD, cLUC-NB, or cLUC was co-expressed with SBP1-nLUC, SBP1$^{RM}$-nLUC, or nLUC, respectively. Luciferase activity was detected at 22 hpi. (C–E) Co-immunoprecipitation analyses of the interaction between SBP1 and SD, NB-LRR or full-length Sw-5b or Sw-5b$^{Heinz}$. SD-FLAG (C), FLAG-NB-LRR (D), and YFP-Sw-5b or YFP-Sw-5b$^{Heinz}$ (E) were assayed for co-immunoprecipitation with SBP1$^{RM}$-RFP or RFP control. Immunoprecipitation was carried out using antibodies specified in the figure and then immunoblot was carried out to detect proteins. IB, immunoblot with specific antibody; IP, immunoprecipitation with specific antibody. The sizes of protein are shown on the left. A ponceau S stained band is shown to serve as the loading control. All experiments were repeated at least three times with similar results. Source data are provided as a Source data file. All experiments were repeated at least three times with similar results. Source data are available online for this figure.

Rpi-blb2 mRNA level (Appendix Fig. S4E,F), suggesting that SBP1 regulates the protein stability of R8 but not Rpi-blb2.

Next, we examined whether *SBP1* knockout affects Sw-5b-mediated immunity. To test this, we used a recently developed reverse-genetics based TSWV GFP reporter system (Feng et al, 2020) in which the GFP expression is correlated with the amount of viral accumulation. We co-expressed Sw-5b with infectious clones of TSWV, L$_{(+)opt}$ + M$_{(-)opt}$ + SR$_{(+)eGFP}$, in WT, *sbp1-1* and *sbp1-2* plant, respectively. As shown in Fig. 3H, co-expression of L$_{(+)opt}$ + M$_{(-)opt}$ + SR$_{(+)eGFP}$ result in high accumulation of TSWV carrying GFP in a large area of WT *N. benthamiana* leaves. When the

Agrobacteria carrying Sw-5b were infiltrated at OD$_{600}$ = 0.25, almost no GFP fluorescence can be detected in both WT and *sbp1-1* mutant plant leaves (Appendix Fig. S6A,B), indicating that the expression of Sw-5b at high level induced very strong defense against TSWV. We then reduced the concentration of Agrobacteria carrying Sw-5b to OD$_{600}$ = 0.1, this allowed TSWV-GFP to accumulate at a low level. In *sbp1-1* and *sbp1-2* plant, co-expression of Sw-5b at OD$_{600}$ = 0.1 further reduced GFP accumulation of TSWV replicons to much lower levels compared to those in WT plant (Fig. 3H,I). These data show that knocking out *SBP1* enhances the Sw-5b-mediated resistance to TSWV.

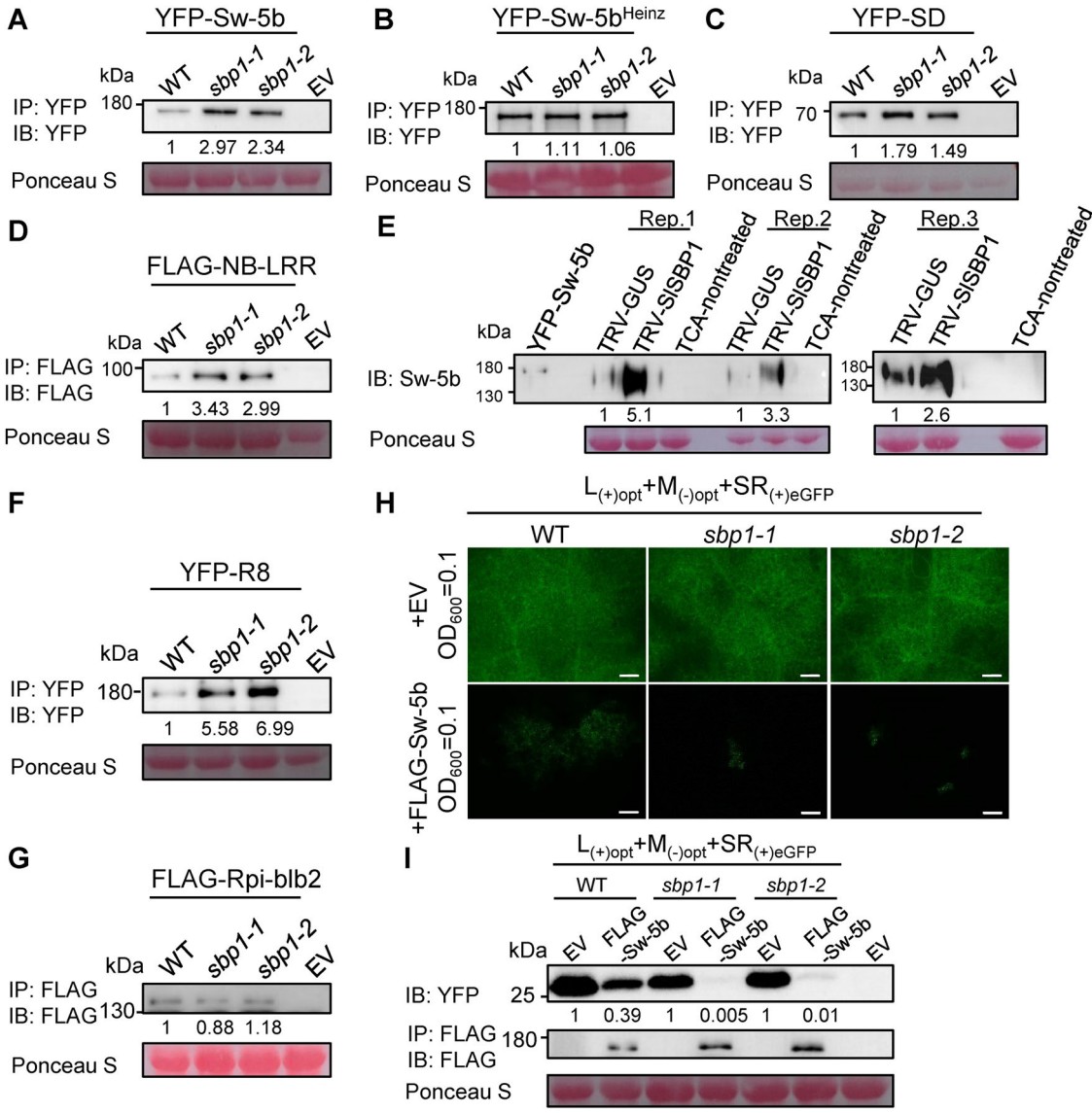

**Figure 3. E3 ligase SBP1 regulates homeostasis of Sw-5b and Sw-5b-mediated immunity.**

(A–D) Knock out of *SBP1* significantly enhances protein accumulation of Sw-5b, SD, and NB-LRR but has no effect on Sw-5b$^{Heinz}$ in *N. benthamiana* plant leaves. YFP-Sw-5b (A), YFP-Sw-5b$^{Heinz}$ (B), YFP-SD (C), and FLAG-NB-LRR (D) were expressed in WT, *sbp1-1* or *sbp1-2* mutant *N. benthamiana* leaves. Total protein was extracted from the samples 24 hpi and analyzed by immunoblot. (E) Knocking down *SlSBP1* enhances the protein accumulation level of Sw-5b in tomato carrying *Sw-5b* gene. Total protein extracted from TRV-GUS or TRV-*Sl*SBP1 tomato leaves were precipitated by Trichloroacetic acid (TCA)-Acetone. Total protein extracted from TRV-GUS tomato leaves without TCA-Acetone precipitation was used as a control (TCA-nontreated). The blot was detected by Sw-5 b-specific antibodies. The YFP-Sw-5b expressed in *N. benthamiana* leaves was used as a positive control to indicate that Sw-5b antibodies are functional. Rep. 1-3 refer to three independent repeats of the experiments. (F, G) Protein accumulation of R8 and Rpi-blb2 in SBP1-KO *N. benthamiana*. YFP-R8 (F) and FLAG-Rpi-blb2 (G) were transiently expressed in WT, *sbp1-1* or *sbp1-2* mutant *N. benthamiana* plant leaves and protein accumulation were analyzed by immunoblot using YFP or FLAG-specific antibodies at 30 hpi. In (A) to (F), IB, immunoblot with specific antibody; IP, immunoprecipitation with specific antibody. The sizes of protein are shown on the left. A ponceau S-stained band is shown to serve as the loading control. Protein accumulation level was quantified by ImageJ software. (H, I) The infectious clones of TSWV carrying GFP reporter (final OD$_{600}$ of 0.25) was co-expressed with Sw-5b or EV (final OD$_{600}$ of 0.1) in WT, *sbp1-1* or *sbp1-2* mutant plant leaves via agrobacterium. GFP fluorescence and protein accumulation were detected by inverted fluorescence microscopy (H) or Western blot using YFP-specific antibodies (I) at 2 days post infiltration (dpi). The EV was used as a control. Bar = 100 μm. Protein accumulation level was quantified by ImageJ software. IB, immunoblot with specific antibody; IP, immunoprecipitation with specific antibody. Ponceau S staining was used to show the amount of protein loaded. Source data are provided as a Source data file. All experiments were repeated at least three times with similar results. Source data are available online for this figure.

## SBP1 is capable of ubiquitinating Sw-5b

We next investigated whether SBP1 displays ubiquitin ligase activity towards Sw-5b. To test this, we performed an in vitro

ubiquitination assay. The UBA1 (E1), UBC8 (E2), SBP1-HA-His, His-FLAG-Ub, and GST-Sw-5b proteins were expressed and purified from *E. coli*. The purified proteins were mixed and assayed for its ubiquitination in vitro. The ubiquitination bands were

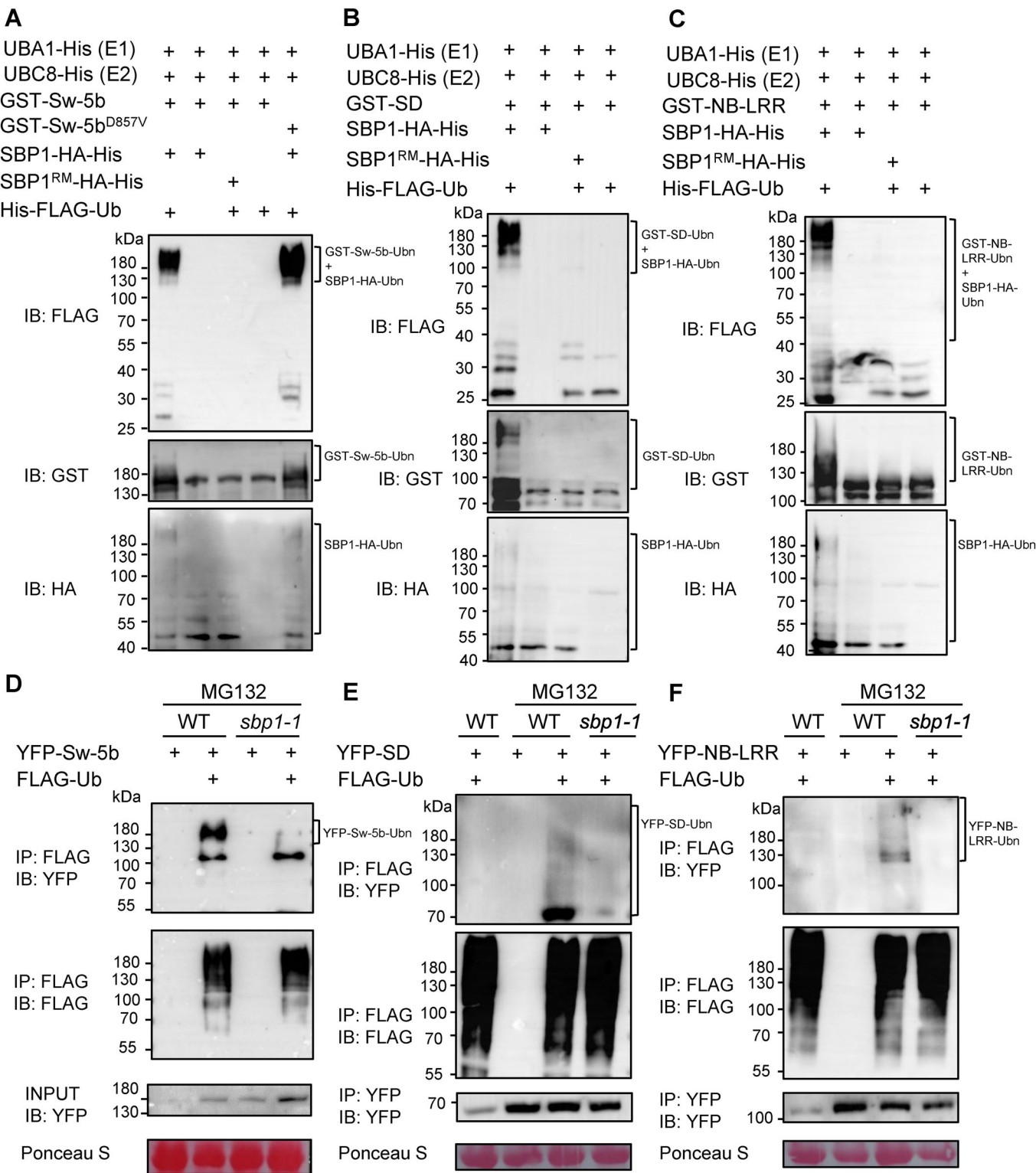

examined using an immunoblot. When probed with antibodies to detect FLAG-Ub, high-molecular-weight bands were detected in samples where GST-Sw-5b was treated with E1, E2, SBP1-HA-His and His-FLAG-Ub (Fig. 4A), indicating that Sw-5b was poly-ubiquitinated by SBP1. In the absence of SBP1, no high-molecular-weight bands was detected for GST-Sw-5b (Fig. 4A). Moreover, when the LOF SBP1$^{RM}$ mutant was used in place of WT SBP1, no GST-Sw-5b ubiquitination band was detected. We also examined whether Sw-5b SD and NB-LRR can be ubiquitinated by SBP1 using a similar assay. As shown in Figs. 4B,C, both GST-SD and

**Figure 4.   SBP1 is capable of ubiquitinating Sw-5b.**

(A–C) In vitro ubiquitination assay of Sw-5b, Sw-5b$^{D857V}$, SD, and NB-LRR by SBP1. GST-Sw-5b (**A**), GST-Sw-5b$^{D857V}$ (**A**), GST-SD (**B**), or GST-NB-LRR (**C**) was incubated with UBA1-His (E1), UBC8-His (E2) and His-FLAG-Ub without or with SBP1-HA-His or SBP1$^{RM}$-HA-His. All these proteins were expressed and purified from *E. coli*. The proteins were incubated at 26 °C for 2 h and followed by immunoblot analysis using FLAG, GST, or HA-specific antibodies. The sizes of protein are shown on the left. (**D–F**) In vivo ubiquitination assays of Sw-5b, SD, and NB-LRR in wild-type (WT) and *SBP1* knock out *N. benthamiana* plant leaves. YFP-Sw-5b, YFP-SD, or YFP-NB-LRR was co-expressed with FLAG-Ub or EV in leaves of WT or *sbp1-1* mutant *N. benthamiana* plants, 25 μM MG132 was infiltrated into the leaves at 22 hpi. The ubiquitination of Sw-5b (YFP-Sw-5b-Ubn), SD (YFP-SD-Ubn), or NB-LRR (YFP-NB-LRR-Ubn) proteins was detected at 30 hpi by immunoprecipitation using anti-FLAG beads followed by detection with YFP-specific antibodies. The overall ubiquitinated proteins were detected using FLAG-specific antibodies. For all panels: IB, immunoblot with specific antibody; IP, immunoprecipitation with specific antibody. The sizes of protein are shown on the left. A ponceau S-stained band is shown to serve as the loading control. Source data are provided as a Source data file. All experiments were repeated at least three times with similar results. Source data are available online for this figure.

GST-NB-LRR have high-molecular-weight bands when probed with antibodies to detect FLAG-Ub, suggesting that SBP1 can ubiquitinate both Sw-5b SD and NB-LRR.

Next, we performed an in vivo assay to detect Sw-5b ubiquitination by SBP1. We expressed YFP-Sw-5b with or without FLAG-Ub in WT and *sbp1-1* *N. benthamiana*. The total ubiquitinated proteins were immunoprecipitated (IP) using anti-FLAG beads and the IP proteins were probed with YFP-specific antibodies. The results showed that YFP-Sw-5b is ubiquitinated in WT *N. benthamiana* plants, however, this ubiquitination is significantly reduced in the *sbp1-1* mutant (Fig. 4D). We also examined the ubiquitination of Sw-5b$^{Heinz}$ in WT and *SBP1* knockout *N. benthamiana* leaves. The results showed that there was no difference in the ubiquitination of Sw-5b$^{Heinz}$ between WT and *SBP1*-KO *N. benthamiana* leaves (Fig. EV2A). Subsequently, we examined the ubiquitination of Sw-5b SD and NB-LRR domains in WT and *sbp1-1* mutant *N. benthamiana* leaves. As shown in Fig. 4E,F, both the SD and NB-LRR domains were ubiquitinated in WT plants, whereas these ubiquitinations were significantly reduced in the *sbp1-1* plants.

To determine the amino acids within Sw-5b that was ubiquitinated by SBP1, Sw-5b SD and NB-LRR domains ubiquitinated by SBP1 were analyzed by Mass spectrometry. The amino acids K79, K81, and K153 in the SD domain were identified with ubiquitination modification, whereas attempts to identify ubiquitinated amino acids in NB-LRR failed. K79, K81, and K153 in the SD domain were all mutated to arginine (R). The immunoblot results showed that the protein levels of YFP-SD$^{K79/81/153R}$ mutant were significantly increased in WT *N. benthamiana* leaves compared to WT YFP-SD, whereas these differences were not detectable in *SBP1*-KO plants (Fig. EV2B). We have previously identified the small region in LRR encompassing polymorphic sites 3–6 (LRR3-6) that are required for the NSm recognition (Zhu et al, 2017). To investigate whether SBP1 directly regulates the small region encompassing the NSm recognition sites of LRR, we fused the 945–1055 amino acid of Sw-5b or Sw-5b$^{Heinz}$ with YFP at the C-terminus (Fig. EV2C,D) and analyzed their protein levels in both WT and *sbp1-1* plants. As shown in Fig. EV2D, the protein accumulation of Sw-5b$^{945-1055}$-YFP was lower than Sw-5b$^{Heinz945-1055}$-YFP in WT *N. benthamiana* plants, however, their protein levels were similar in *sbp1-1* plants. There are nine lysines in the LRR3-6 region. These nine lysine sites in Sw-5b LRR3-6 were analyzed in four mutants (Sw-5b$^{945-1055/K1-3R}$, Sw-5b$^{945-1055/K4-5R}$, Sw-5b$^{945-1055/K6-7R}$, and Sw-5b$^{945-1055/K8-9R}$), in which K946-K964-K970, K1013-K1014, K1022-K1027 and K1036-K1044 were replaced with arginine (R), respectively. As shown in Fig. EV2E, protein levels of Sw-5b$^{945-1055/K6-7R}$ mutant was significantly higher than that of WT Sw-5b and other mutants, suggesting that amino acid residues K1022 and K1027 are important for the ubiquitination of LRR3-6 by SBP1.

## TSWV NSm disrupts interaction of both Sw-5b SD and NB-LRR domains with SBP1 and interferes with ubiquitination of Sw-5b

From previous experiments, we saw that TSWV NSm protects Sw-5b from 26S-proteasome degradation. To gain insight on the molecular mechanism of how TSWV NSm protects Sw-5b from 26S-proteasome degradation, we co-expressed YFP-Sw-5b and SBP1$^{RM}$-RFP with elicitor TSWV NSm or non-elicitor NSm from TZSV (Zhu et al, 2017) to test whether TSWV NSm could specifically affect the interaction between SBP1 and Sw-5b. As shown in Fig. 5A, YFP-Sw-5b associates with SBP1$^{RM}$-RFP without TSWV NSm (EV control) or with non-elicitor TZSV NSm, however, this interaction is strongly disrupted in the presence of elicitor TSWV NSm. The protein accumulation of SBP1$^{RM}$-RFP was not affected by TSWV NSm (Fig. 5A). Subsequently, we conducted competitive GST pull-down assay to investigate the impact of the TSWV effector NSm on SBP1-Sw-5b interaction. The results showed that GST-Sw-5b exhibited a reduced capacity to bind SBP1 in the presence of elicitor TSWV NSm, but not non-elicitor TZSV NSm, and this effect was dose-dependent (Fig. 5B). Our recent study showed that TSWV NSm interacts with both Sw-5b SD and LRR domains (Li et al, 2019). So, we also examined whether TSWV NSm can disrupt the interaction of SBP1 with Sw-5b SD and with NB-LRR. Co-IP assays showed that the interaction between SBP1 and Sw-5b SD, and between SBP1 and Sw-5b NB-LRR were largely disrupted by the presence of elicitor TSWV NSm but not non-elicitor TZSV NSm (Fig. EV3A,B). Subsequently, we conducted the competitive split-LUC experiments in planta to investigate the impact of the NSm on SBP1-Sw-5b SD/NB-LRR interactions. As illustrated in Fig. EV3C–F, when the OD$_{600}$ of Agrobacterium carrying TSWV NSm was elevated from 0 to 0.5, the interactions between SBP1 and Sw-5b SD or between SBP1 and NB-LRR were markedly inhibited in comparison to the effect of TZSV NSm.

Our previous studies also showed that the 21-amino acid peptide region of TSWV effector NSm (NSm$^{21}$) can directly bind to both SD and NB-LRR and is sufficient to trigger Sw-5b-mediated HR (Li et al, 2019; Zhu et al, 2017). Similar to the full-length NSm, the elicitor TSWV NSm$^{21}$ can also strongly disrupt the interaction between full-length Sw-5b, and its SD and NB-LRR domains with SBP1$^{RM}$, but the non-elicitor TZSV NSm$^{21}$ cannot (Fig. EV3G–I). These data suggest that TSWV NSm is able to specifically disrupt the interaction of SBP1 with both the SD and NB-LRR domains of Sw-5b NLR.

Next, we examined whether viral effector NSm can interfere with the ubiquitination of Sw-5b by SBP1 in vivo and in vitro. As shown in Fig. 5C,D, the addition of TSWV NSm significantly

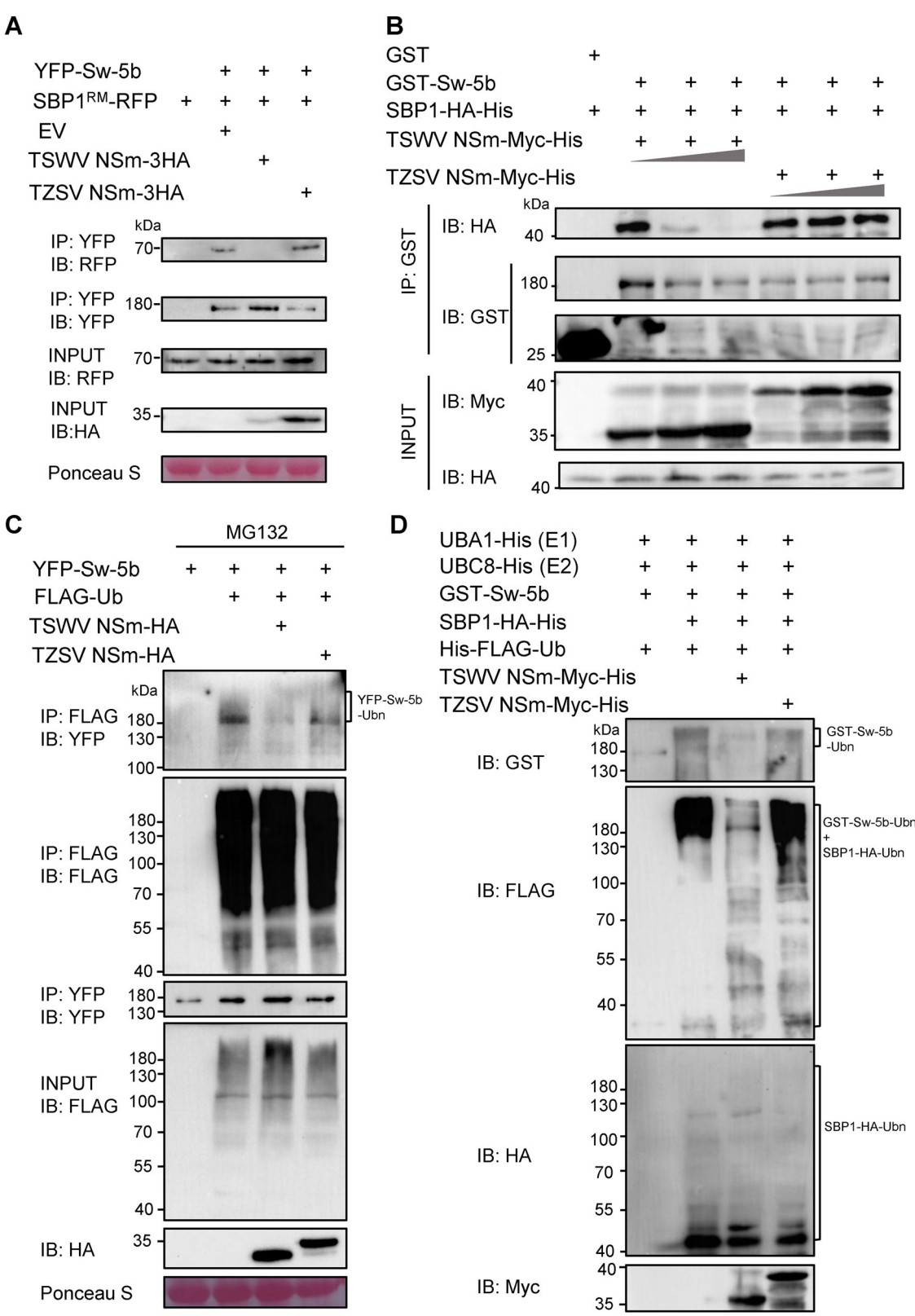

◄

**Figure 5.  Viral NSm disrupts Sw-5b-SBP1 interaction and interferes with the ubiquitination of Sw-5b by SBP1.**

(A) The inhibitory effects of TSWV NSm on the interaction of Sw-5b with SBP1 in vivo. YFP-Sw-5b was immunoprecipitated with SBP1$^{RM}$-RFP in the presence of EV, TSWV NSm, or TZSV NSm. SBP1$^{RM}$-RFP alone was also used as a control. Samples were harvested 21 hpi and blots were detected using YFP, FLAG, RFP, and HA-specific antibodies. (B) The competitive GST pull-down assay for the inhibitory effects of TSWV NSm on the Sw-5b-SBP1 interaction. GST-Sw-5b was used to pull-down SBP1-HA-His with increasing amounts of purified TSWV NSm-Myc or TZSV NSm-Myc. GST was added as a control. The blots were probed with GST, HA, and Myc-specific antibodies. (C) In vivo interference analysis of TSWV NSm or TZSV NSm on the ubiquitination of Sw-5b in *N. benthamiana* leaves. YFP-Sw-5b and TSWV NSm or YFP-Sw-5b and TZSV NSm were co-expressed with FLAG-Ub in *N. benthamiana* leaves, 25 μM MG132 was infiltrated into the leaves at 16 hpi. The ubiquitinated proteins were immunoprecipitated at 24 hpi using anti-FLAG beads. The ubiquitination of Sw-5b proteins (YFP-Sw-5b-ubn) was detected using YFP-specific antibodies. The overall ubiquitinated proteins were detected using FLAG-specific antibodies. (D) In vitro interference assays of TSWV NSm or TZSV NSm on the ubiquitination of Sw-5b (GST-Sw-5b-ubn). GST-Sw-5b proteins were incubated with UBA1-His (E1), UBC8-His (E2), SBP1-HA-His and His-FLAG-Ub with TSWV NSm-Myc-His or TZSV NSm-Myc-His. All of the aforementioned proteins were expressed and purified from *E. coli*. For the in vitro ubiquitination analysis, the proteins were incubated at 26 °C for 2 h and followed by immunoblot analysis using FLAG, GST, HA, or Myc-specific antibodies. In (A) to (D), the size of protein was shown on the left. IB, immunoblot with specific antibody; IP, immunoprecipitation with specific antibody. Ponceau S staining was used to show the amount of protein loaded. Source data are provided as a Source data file. All experiments were repeated at least three times with similar results. Source data are available online for this figure.

diminished the ubiquitination of Sw-5b, however, the addition of TZSV NSm did not have such effect. Furthermore, the addition of TSWV NSm neither influenced the overall ubiquitination of plant endogenous proteins in vivo nor the ubiquitination of SBP1 itself in vitro (Fig. 5C,D).

## The stability of inactive state Sw-5b is regulated by its SD domain while the active state Sw-5b protein level is regulated by both its SD and LRR domains

As the non-resistant Sw-5b$^{Heinz}$ homolog cannot recognize NSm and its protein stability is not affected by NSm, we examined the differences in the protein homeostasis of the resistant Sw-5b and its non-resistant Sw-5b$^{Heinz}$ homolog. Compared to the non-resistant YFP-Sw-5b$^{Heinz}$, YFP-Sw-5b accumulated less (Fig. EV4A). The difference is not due to the mRNA transcript difference of Sw-5b and Sw-5b$^{Heinz}$ (Fig. EV4B).

The findings, that both SD and LRR were degraded by 26S proteasome pathway, raise a question as to why Sw-5b needed to evolve two domains to be targeted by SBP1 and both were protected by elicitor TSWV NSm. We previously showed that in the absence of TSWV NSm, the SD, CC and NB-LRR domains interact with each other to maintain Sw-5b in an inactive state. In the presence of TSWV NSm, however, suppression by the SD, CC and LRR is sequentially relieved, and Sw-5b switches to an active state (Chen et al, 2016). We previously also identified that Sw-5b SD domain and polymorphic sites 3 to 6 in the Sw-5b LRR domain, that show amino acid differences compared to the non-resistant Sw-5b$^{Heinz}$ homolog (Fig. EV2C), are critical for NSm recognition (Zhu et al, 2017; Li et al, 2019). To test whether the "ON/OFF states" of the Sw-5b protein may have a differential need for these two domains to maintain their protein level, we first created two chimeric proteins of Sw-5b by swapping into it either the previously described SD domain [SD$^{Heinz}$-(CC-NB-LRR)$^{Sw-5b}$] or the LRR polymorphic sites 3–6 (Sw-5b$^{LRRM3-6}$) (Fig. EV4C) from the non-resistant Sw-5b$^{Heinz}$. The immunoblot assay showed that SD$^{Heinz}$-(CC-NB-LRR)$^{Sw-5b}$ accumulated to much higher levels compared to WT Sw-5b. However, Sw-5b$^{LRRM3-6}$ protein level was similar to that of WT Sw-5b (Fig. EV4D). Further, we observed that SD$^{Heinz}$-(CC-NB-LRR)$^{Sw-5b}$ can induce slight cell death (Appendix Fig. S7A). These data suggest that the protein level of the inactive state of full-length Sw-5b is regulated only by its SD domain and not its LRR since swapping out Sw-5b's LRR

polymorphic sites 3–6 from Sw-5b$^{Heinz}$ did not lead to enhanced protein accumulation.

Next, we exchanged the SD domain and LRR polymorphic sites 3–6 from Sw-5b$^{Heinz}$ into Sw-5b$^{D857V}$ (Fig. EV4E), a previously described auto-active mutant with a D857V mutation in the conserved MHD motif (Chen et al, 2016). Both SD$^{Heinz}$-(CC-NB-LRR)$^{Sw-5b/D857V}$ with SD domain from Sw-5b$^{Heinz}$ and Sw-5b$^{D857V/LRRM3-6}$ with LRR polymorphic sites 3–6 from Sw-5b$^{Heinz}$ induced constitutive HR cell death in *N. benthamiana* leaves (Appendix Fig. S7B,C,E). The immunoblot results showed that SD$^{Heinz}$-(CC-NB-LRR)$^{Sw-5b/D857V}$ with SD domain from Sw-5b$^{Heinz}$, Sw-5b$^{D857V/LRRM3-6}$ with polymorphic sites 3–6 from Sw-5b$^{Heinz}$, or SD$^{Heinz}$-CNL$^{D857V/LRRM3-6}$ with both SD and polymorphic sites 3–6 of LRR from Sw-5b$^{Heinz}$ accumulated higher protein levels than Sw-5b$^{D857V}$ (Fig. EV4F; Appendix Fig. S7D,E), suggesting that the protein stability of the active state Sw-5b is regulated by both SD and LRR.

Next, we examined the ubiquitination of the above chimeric Sw-5b variants in WT and *SBP1* knockout *N. benthamiana* leaves. As shown in Fig. EV4G, the ubiquitination of YFP-Sw-5b and YFP-Sw-5b$^{LRRM3-6}$ was higher than that of YFP-SD$^{Heinz}$-(CC-NB-LRR)$^{Sw-5b}$. However, the ubiquitination of these chimeric Sw-5b variants was significantly reduced in the *SBP1* knockout mutant. Moreover, the ubiquitination of YFP-Sw-5b$^{D857V}$ was found to be higher than that of YFP-SD$^{Heinz}$-CNL$^{Sw-5b/D857V}$ and YFP-Sw-5b$^{D857V/LRRM3-6}$. Similarly, the ubiquitination of these chimeric Sw-5b variants was significantly reduced in *SBP1* knockout plants (Fig. EV4G).

## Protein turnover rate of the active form Sw-5b is much higher than for its inactive form

The above results showed that both SD and LRR domains are involved in maintaining the homeostasis of the active form Sw-5b. Since the active form of NLR is the main reason for high fitness cost for plants, we hypothesized that the active form of Sw-5b might be degraded faster than its inactive form due to the additive effect of both the SD and LRR domains being targeted for degradation by the 26S proteasome. To test this hypothesis, YFP-Sw-5b and YFP-Sw-5b$^{D857V}$ were transiently expressed in *N. benthamiana* leaves. At 20 h post infiltration, CHX was infiltrated into the leaves expressing YFP-Sw-5b and YFP-Sw-5b$^{D857V}$ to arrest protein translation. Protein levels of YFP-Sw-5b and YFP-Sw-5b$^{D857V}$ were examined 0, 1, 2 to 3 h post CHX treatment. As shown in Fig. 6A,B, the YFP-Sw-5b$^{D857V}$ protein degraded much faster compared to YFP-Sw-5b, suggesting that the active form of Sw-5b has a high turnover rate.

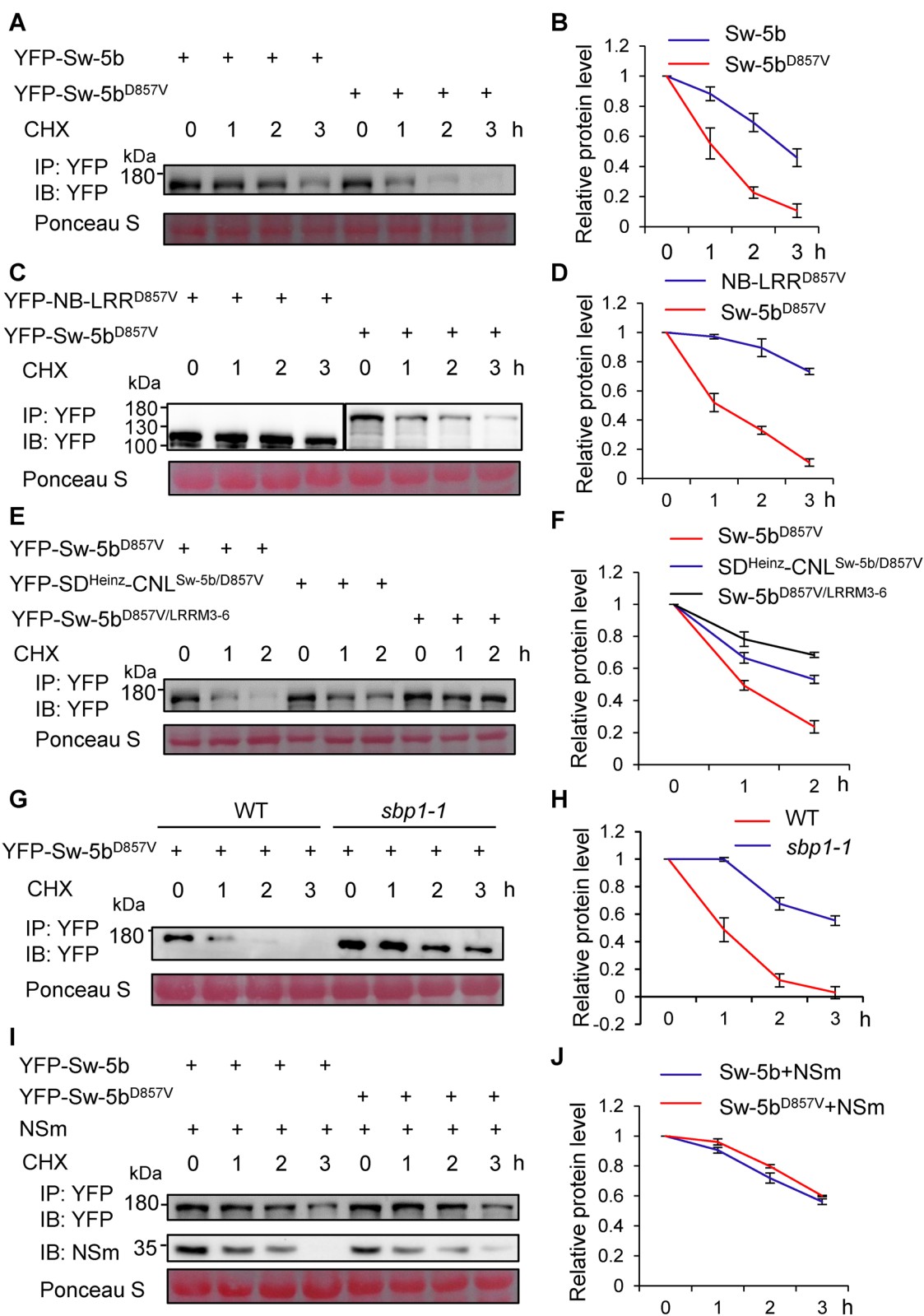

We also performed an in vitro time-course degradation assay for the Sw-5b and Sw-5b$^{D857V}$. GST-Sw-5b and GST-Sw-5b$^{D857V}$ expressed and purified from *E. coli* were mixed with plant protein extracts and their proteins levels were monitored over 3 h. The results showed that GST-Sw-5b$^{D857V}$ also degraded faster than GST-Sw-5b from 0 to 3 h (Appendix Fig. S8A,B).

To test whether the fast turnover of active form of Sw-5b was because both its SD and LRR domains could be targeted for

◀ **Figure 6. Dynamic homeostasis of the inactive and the active state Sw-5b regulated by SBP1 and viral NSm.**

(A, B) Protein turnover rate of Sw-5b and Sw-5b$^{D857V}$ in planta. YFP-Sw-5b or YFP-Sw-5b$^{D857V}$ was transiently expressed in *N. benthamiana* leaves and treated with 10 μg/mL cycloheximide (CHX) at 20 hpi to block protein synthesis. Samples were taken at 0–3 h after CHX treatment. The samples were analyzed by immunoblotting using YFP-specific antibodies. Protein accumulation level in panel (**A**) was quantified by ImageJ and was shown in panel (**B**). (C, D) Protein turnover rate of Sw-5b NB-LRR$^{D857V}$ and full-length Sw-5b$^{D857V}$ in planta. The assay was performed as indicated in (**A**). Protein accumulation level in panel (**C**) was quantified by ImageJ and was shown in panel (**D**). (E, F) Protein turnover rate of YFP-Sw-5b$^{D857V}$, YFP-SD$^{Heinz}$-(CC-NB-LRR)$^{Sw-5b/D857V}$ [YFP-SD$^{Heinz}$-CNL$^{Sw-5b/D857V}$], and YFP-Sw-5b$^{D857V/LRRM3-6}$ in planta. The assay was performed as indicated in (**A**). Protein level was detected at the indicated time. Protein accumulation level in panel (**E**) was quantified by ImageJ software and was shown in panel (**F**). (G, H) Protein turnover rate of Sw-5b$^{D857V}$ in WT or *sbp1-1* mutant line of *N. benthamiana* plant leaves. Protein accumulation level was quantified and shown in panel (**H**). (I, J) Protein turnover rate of Sw-5b vs Sw-5b$^{D857V}$ in the presence of TSWV NSm in WT *N. benthamiana* leaves. YFP-Sw-5b or YFP-Sw-5b$^{D857V}$ was co-expressed with TSWV NSm in *N. benthamiana* leave and treated with 10 μg/mL CHX at 20 hpi. Protein accumulation level was quantified and shown in panel (**J**). In (**B**), (**D**), (**F**), (**H**), and (**J**), data are presented as means ± SD (*n* = 3 biologically independent samples). In (**A**) to (**F**), the sizes of protein are shown on the left. IB, immunoblot with specific antibody; IP, immunoprecipitation with specific antibody. A ponceau S stained band is shown to serve as the loading control. Protein accumulation level was quantified by Image J software. Source data are provided as a Source data file. All experiments were repeated at least three times with similar results. Source data are available online for this figure.

degradation, we used a previously characterized auto-active mutant NB-LRR$^{D857V}$ without SD-CC domain (Chen et al, 2016). This NB-LRR$^{D857V}$ mutant induced HR cell death in the absence of TSWV NSm in *N. benthamiana* leaves (Appendix Fig. S8C–F). We analyzed protein turnover rate of the full-length Sw-5b$^{D857V}$ and this mutant without SD-CC domain using an in planta time-course degradation assay as described above. Results showed that the auto-active Sw-5b$^{D857V}$ has significantly faster degradation rate compared to the auto-active NB-LRR$^{D857V}$ (Fig. 6C,D).

Next, we compared the turnover rate of Sw-5b$^{D857V}$ with that of SD$^{Heinz}$-(CC-NB-LRR)$^{Sw-5b/D857V}$ [SD$^{Heinz}$-CNL$^{Sw-5b/D857V}$] and Sw-5b$^{D857V/LRRM3-6}$. As shown in Fig. 6E,F, both YFP-SD$^{Heinz}$-CNL$^{Sw-5b/D857V}$ and YFP-Sw-5b$^{D857V/LRRM3-6}$ protein degrades slower than YFP-Sw-5b$^{D857V}$. Taken together, we conclude that the additive effect of the SD and NB-LRR as targets for protein degradation is responsible for the rapid turnover of the active state Sw-5b.

## Homeostasis of active state Sw-5b is regulated by the E3 ligase SBP1 and the TSWV effector NSm

Next, we examined whether SBP1 plays a role in the rapid protein turnover of the active state of Sw-5b. As shown in Fig. 6G,H, Sw-5b$^{D857V}$ expressed in WT *N. benthamiana* leaves was rapidly turned over *in planta* within 3 h of CHX treatment. However, the rate of Sw-5b$^{D857V}$ protein turnover was significantly reduced in the *sbp1-1* mutant. The protein turnover of the WT Sw-5b, chimeric SD$^{Heinz}$-CNL$^{Sw-5b/D857V}$ and Sw-5b$^{D857V/LRRM3-6}$ mutants was slower than that of the Sw-5b$^{D857V}$ in the WT *N. benthamiana* leaves. Furthermore, the protein turnover rate of these Sw-5b variants was reduced in the *sbp1-1* mutant (Fig. EV5A–C). We also examined GST-Sw-5b$^{D857V}$ ubiquitination by SBP1 in the in vitro ubiquitination assay. GST-Sw-5b$^{D857V}$ expressed and purified from *E. coli* was mixed with the purified E1, E2, SBP1 and FLAG-Ub proteins. The pulled down GST-Sw-5b$^{D857V}$ displayed strong high-molecular-weights bands in the presence of SBP1 and FLAG-Ub (Fig. 4A), suggesting that the auto-active state of Sw-5b is strongly ubiquitinated by SBP1. To investigate the expression of *SBP1* during the activation of Sw-5b-mediated immunity, a time-course analysis of *SBP1* mRNA levels was conducted on WT *N. benthamiana* plants expressing Sw-5b or Sw-5b$^{D857V}$. In parallel, we examined *SBP1* mRNA levels in WT plants co-expressing Sw-5b and TSWV NSm and used plants co-expressing Sw-5b and non-elicitor TZSV NSm as a control. The results showed that the *SBP1* mRNA expression levels were

upregulated during the activation of Sw-5b-mediated immunity (Fig. EV5D,E). Subsequently, a time-course analysis of conductivity was conducted on WT and *sbp1-1* mutant plants expressing Sw-5b$^{D857V}$ or co-expressing Sw-5b and TSWV NSm. The results showed that the conductivity in *sbp1-1* mutant plants treated with Sw-5b$^{D857V}$ was higher than that in WT plants at 24–30 hpi and that there was no significant difference in conductivity between WT and *sbp1-1* plants at 36–48 hpi (Fig. EV5F left panel). However, there was no discernible difference in conductivity induced by Sw-5b and TSWV NSm in WT and *sbp1-1* plants from 24 to 48 hpi (Fig. EV5F right panel).

Next, we examined the effect of TSWV NSm on the protein turnover rate of Sw-5b$^{D857V}$. To test this, we performed an in planta time-course degradation assay for Sw-5b and Sw-5b$^{D857V}$ in the presence of TSWV effector NSm. The results show that YFP-Sw-5b and YFP-Sw-5b$^{D857V}$ gradually degraded from 0 to 3 h. However, the degradation rate of YFP-Sw-5b$^{D857V}$ in the presence of TSWV NSm was nearly the same as that of YFP-Sw-5b in the presence of TSWV NSm (Fig. 6I,J), suggesting that the protein turnover rate of the active-state Sw-5b is significantly reduced in the presence of viral effector NSm. Moreover, transient over-expression of SBP1 inhibited cell death induced by Sw-5b$^{D857V}$ (Fig. EV5G,H). However, SBP1 had no effect on Sw-5b$^{D857V}$ induced cell death in *N. benthamiana* leaves co-expressing NSm (Fig. EV5G,H), suggesting TSWV effector NSm antagonizes the effect of SBP1 on Sw-5b.

## Discussion

In this study, we demonstrated that the homeostasis of "ON/OFF state" of Sw-5b NLR is antagonistically and dynamically controlled by E3 ligase SBP1 and TSWV effector NSm to induce robust immunity. We found that Sw-5b NLR protein is tightly regulated by the 26S proteasome degradation pathway and is maintained in low levels in the absence of TSWV NSm. The E3 ligase SBP1 was found to directly interact with both Sw-5b N-terminal *Solanaceae* domain (SD) and NB-LRR domains and is required for the regulation of Sw-5b homeostasis. The E3 ligase SBP1 was upregulated at the transcript level upon the activation of Sw-5b, which leads to fast protein turnover of the active state of Sw-5b. We found that TSWV NSm that is recognized by Sw-5b, competes with SBP1 for interaction with both Sw-5b SD and NB-LRR, thereby stabilizing

the active form of Sw-5b NLR to accumulate to a sufficient level to trigger robust defense responses.

E3 ubiquitin ligase SBP1 was found to interact with Sw-5b NLR but not with Sw-5b$^{Heinz}$ homolog. Knockout of *SBP1* in *N. benthamiana* led to enhanced accumulation of Sw-5b NLR but not the non-resistant Sw-5b$^{Heinz}$ homolog. Further ubiquitination assay demonstrates that this is due to the fact that SBP1 specifically ubiquitinates Sw-5b, but not the Sw-5b$^{Heinz}$ homolog. We previously found that both Sw-5b SD and NB-LRR domains recognize and directly interact with TSWV NSm (Zhu et al, 2017; Li et al, 2019). In this study, replacing both SD and polymorphic sites 3–6 of LRR domain of Sw-5b that are important for NSm recognition with the corresponding region from Sw-5b$^{Heinz}$ resulted in higher protein accumulation of the chimeric Sw-5b variants. Both Sw-5b SD and NB-LRR were also found to directly interact with the E3 ligase SBP1. Indeed, the TSWV NSm was able to compete the interaction of SBP1 with both Sw-5b SD and NB-LRR leading to increased accumulation of Sw-5b protein. In addition, TSWV NSm has also been found to interfere with the ubiquitination of Sw-5b. It is noteworthy that the TZSV NSm, the non-elicitor of Sw-5b, does not influence the accumulation of Sw-5b and does not impede the interaction between Sw-5b and SBP1, nor the ubiquitination of Sw-5b mediated by SBP1. These findings indicate that the accumulation stability of the Sw-5b is specifically regulated by its effector. Many studies have revealed that pathogen effectors were able to target E3 ligases (Bos et al, 2010; Park et al, 2012; Ishikawa et al, 2014; Park et al, 2016). However, we found that TSWV NSm did not affect the E3 ligase activity of SBP1 itself. Collectively, these data strongly suggest that TSWV effector NSm competes with the E3 ligase SBP1 for the same binding sites on Sw-5b. SBP1 was also found to regulate the protein stability of potato NLR R8, but not potato NLR Rpi-blb2. Interestingly, Sw-5b and R8 have the same polymorphic site 3–6 difference on this small LRR region (Appendix Fig. S9), suggesting that those polymorphic sequences in the Sw-5b and R8 might be targeted by SBP1. In a recent study, TMV helicase was found to interfere with the interaction between tobacco NLR N and the E3 ligase UBR7 (Zhang et al, 2019), suggesting that effector binding induced stabilization of NLR proteins may be a widespread phenomenon.

We previously showed that in the absence of TSWV NSm, Sw-5b LRR inhibits NB-ARC, and SD and CC further additively inhibit NB-LRR, thereby maintaining Sw-5b in a multilayered autoinhibited state (Chen et al, 2016). In the presence of viral NSm, SD, CC, and LRR inhibitions were sequentially relieved from NB-ARC, leading to the switch of Sw-5b from an inactive state to an active state (Chen et al, 2016). Arabidopsis NLR ZAR1 was found to undergo major conformational changes from the monomer in inactive state to the pentamer in active state (Wang et al, 2019; Ma et al, 2020). When Sw-5b is in the multilayered auto-inhibition state, the LRR domain may not be accessible to SBP1. Although SBP1 interacts with both SD and NB-LRR, SBP1 E3 ligase may only target Sw-5b SD when Sw-5b is in the multilayered autoinhibited state. However, when inactive Sw-5b switches into the active form, it undergoes a conformational change that allow the release of both SD and LRR, rendering them accessible to SBP1 for degradation. This is supported by the findings that SD$^{Heinz}$-(CC-NB-LRR)$^{Sw-5b}$, in which the SD of Sw-5b$^{Heinz}$ was swapped into Sw-5b NLR, accumulates to higher levels than Sw-5b. However, Sw-5b$^{LRRM3-6}$, in which the polymorphic sites 3–6 from LRR domain of Sw-5b$^{Heinz}$

was swapped into Sw-5b, had similar protein accumulation level as Sw-5b. But when auto-active Sw-5b$^{D857V}$ was used in a similar assay, both SD$^{Heinz}$-(CC-NB-LRR)$^{Sw-5b-D857V}$ and Sw-5b$^{D857V/LRRM3-6}$ had higher protein accumulation than Sw-5b$^{D857V}$. These findings suggest that both SD and NB-LRR domains of the active Sw-5b are targeted by SBP1 E3 ligase and this leads to fast degradation of the active Sw-5b. This is further supported by the finding that Sw-5b$^{D857V}$ degraded faster than NB-LRR$^{D857V}$, an constitutive mutant without SD-CC, and the finding that both SD$^{Heinz}$-(CC-NB-LRR)$^{Sw-5b/D857V}$ and Sw-5b$^{D857V/LRRM3-6}$ chimeric proteins degraded slower than Sw-5b$^{D857V}$. It is interesting that the self-active mutant variants of Sw-5b [SD$^{Heinz}$-(CC-NB-LRR)$^{Sw-5b-D857V}$ or Sw-5b$^{D857V/LRRM3-6}$] did not cause stronger cell death than Sw-5b$^{D857V}$, although their stability was increased. This is probably because the conformation of chimeric Sw-5b$^{D857V}$ mutant itself also affect their ability to induce cell death.

Bernoux and colleagues recently suggested a model that the inactive and active form of an NLR are in equilibrium (Bernoux et al, 2016). The equilibrium allows a small number of active-state NLR receptors in the absence of pathogen effector. Binding of pathogen effector causes the equilibrium to shift from the inactive state to active state, leading to the accumulation of large number of active-state NLRs to induce an ETI immune response. This model is consistent with the fact of fitness cost caused by introduction of NLRs into crops through breeding project (Tian et al, 2003). High protein accumulation of NLRs may result in a higher amount of protein in the ATP-bound, active state that causes fitness cost in plants. Previous study showed that co-expression of Sw-5b with P19 protein, a RNA silencing suppressor, induce the cell death in the absence of TSWV NSm (De Oliveira et al, 2016), suggesting that high protein accumulation of Sw-5b may cause the equilibrium shifting from the inactive state to the active state. In this study, we demonstrated that the active form of Sw-5b NLR is degraded much faster than its inactive form. Fast degradation of the active form of Sw-5b is through ubiquitination of both SD and NB-LRR domain by SBP1. Furthermore, our results showed that the activation of Sw-5b is accompanied by an elevated level of *SBP1* transcript accumulation, which further enhances the protein turnover rate of the active form of Sw-5b. However, TSWV NSm that is recognized by Sw-5b prevents the E3 ligase SBP1 from ubiquitinating the SD and LRR domains. Therefore, the presence of TSWV effector NSm stabilizes the active form of Sw-5b and prevents its degradation, which in turn induces a robust defense against TSWV. Based on these findings, we propose that in the absence of pathogen effector, the homeostasis of the inactive NLR is regulated by the E3 ligase SBP1 to maintain it in a low level and the active state of NLR in the equilibrium is rapidly degraded by plant 26S proteasome pathway. However, upon recognition of pathogen effector, the active state of NLR is stabilized by pathogen effector and accumulates to a sufficient level to trigger a robust immunity.

In summary, we demonstrate a dynamic regulation of homeostasis of an intracellular immune receptor, the tomato NLR Sw-5b, in the absence or the presence of viral pathogen effector. We propose a working model for this dynamic regulation of Sw-5b by the E3 ligase SBP1 and viral effector NSm (Fig. 7). In this model, the inactive state and the active state of Sw-5b NLR are in an equilibrium. In the absence of viral pathogen effector NSm, the inactive form of the Sw-5b NLR is targeted via its SD domain by SBP1 and is turned over slowly. However, the active form of the Sw-5b NLR is targeted via both its SD and LRR by SBP1 and SBP1 is upregulated during the activation of Sw-

**Figure 7. A model for the dynamic homeostasis regulation of inactive and active state Sw-5b NLR.**

In the absence of TSWV effector NSm, Sw-5b is maintained in a dynamic equilibrium of inactive state and active state. The equilibrium allows small number of active state Sw-5b. SBP1 interacts with the SD domain of inactive Sw-5b and mediates its degradation. However, the active state of Sw-5b can be recognized by SBP1 through both its SD and LRR domains and SBP1 is upregulated during the activation of Sw-5b, both leading to fast degradation of the active state of Sw-5b. Upon recognition of its cognate viral effector NSm, the equilibrium is shifted from the inactive to the active state. Viral NSm is able to prevent the binding of SBP1 and stabilizes the active state of Sw-5b, leading to protein accumulation of active state Sw-5b to trigger a robust disease resistance. The rate of forward/reverse reactions of the equilibrium are schematically represented by different arrow weights. Arrow with higher weight indicates higher rate and arrow with lower weight indicate lower rate. Dashed arrows indicate the direction of the reaction not unfirmed. Poly-Ub, poly-ubiquitination.

5b, both leading to the fast degradation of the active form of Sw-5b. However, upon recognition of viral effector NSm, the active Sw-5b is stabilized and can no longer bind SBP1 which leads to Sw-5b protein accumulation, resulting in a robust immune response. Given the switch from the inactive state to the active state is a common feature for the members of the large family of plant NLRs, it will be interesting to explore whether other members of plant NLRs are also rapidly degraded in their active state and whether those NLR molecules in an active state are stabilized upon recognition of pathogen effectors leading to triggering a robust immune response.

## Methods

### Plasmid construction

Plasmids p2300S-Sw-5b, p2300S-Sw-5b$^{D857V}$, p2300S-Sw-5b$^{Heinz}$, p2300S-NB-LRR, and p2300S-NSm were described previously

(Feng et al, 2016; Chen et al, 2016; Zhu et al, 2017; Li et al, 2019). All other Sw-5b, Sw-5b$^{Heinz}$ and NB-LRR plasmids used in this study were cloned in the pCambin2300S binary vector under 35S promoter. A 1×FLAG or YFP tag was fused to the N terminus of Sw-5b, NB-LRR, Sw-5b$^{Heinz}$, R8, Rpi-blb2, and other derivatives for immunoblotting or co-immunoprecipitation assays. A 3×FLAG was fused to the C-terminus of Sw-5b SD. The 945–1055 aa fragment of Sw-5b or its mutants were cloned into pBinPLUS-YFP binary vector under 35S promoter. Sw-5b and NB-LRR mutants, and RFP fusions were generated by two-step overlap-extension PCR as described (Zhu et al, 2017; Li et al, 2019). Sw-5b, Sw-5b$^{D857V}$, Sw-5b SD, Sw-5b NB-LRR and other genes were cloned into the pGEX-2TK vector and tagged with a GST tag at the N terminus for *E. coli* expression. NSm-6xHis was cloned in the pET28a vector as described (Li et al, 2019). Sw-5b SD and Sw-5b NB domain were cloned into pGBKT7 vectors to perform yeast two-hybrid assay. To clone the full-length cDNA of SBP1 for split-luciferase complementation assay, immunobolotting, Y2H and other derivative

assays, isolated mRNA was reverse transcribed using oligo d(T) primers to obtain the cDNA. This cDNA was used to amplify the desired genes using PCR with specific primers. The resulting PCR products were cloned into pGBKT7 vectors for Y2H assay, into pCambia1300-n/cLUC vector for luciferase assay, and into pCambia2300S vectors for expression in *N. benthamiana*. To generate pGBK01/Cas9-SBP1 constructs, sgRNA designed by CRISPR-P tool (http://cbi.hzau.edu.cn/crispr/) (Lei et al, 2014) was cloned into pGBK01/Cas9 vector (Fu et al, 2018). To generate various virus induced gene silencing (VIGS) constructs, a 300 bp cDNA fragment of each gene was selected based on SNG-VIGS tool (https://vigs.solgenomics.net/), and cloned into pTRV2 vector (Liu et al, 2002). Primers used in this study are listed in Appendix Table S1.

## Transient expression in *N. benthamiana*

*Agrobacterium tumefaciens* strain GV3101 harboring different expression constructs were infiltrated into 6-week-old *N. benthamiana* leaves as described previously (Zhu et al, 2017). For co-infiltration assays, the bacteria were adjusted to $OD_{600} = 0.25$ in induction medium (10 mM 2-(N-morpholino) ethanesulfonic acid [pH 5.6], 10 mM $MgCl_2$, and 150 mM acetosyringone). *N. benthamiana* plants were grown in a climate controlled chamber with 16 h light and 8 h photoperiod at 25 °C.

## Antibody preparation

For the detection of NSm, RFP, and Sw-5b proteins, a 6×His tag was fused with the proteins, expressed in *E. coli* Rosseta (DE3) cells and purified using Ni-NTA agarose (Qiagen). The purified NSm, RFP, or Sw-5b protein was injected into rabbits to produce polyclonal antibodies.

## Immunoblot and immunoprecipitation assay

Immunoblot and immunoprecipitation assays were performed as previously described (Zhu et al, 2017; Li et al, 2019). Briefly, total protein was extracted from 1.0 g *N. benthamiana* leaves in 2 ml of extraction buffer (25 mM Tris–HCl [pH 7.5], 1 mM EDTA, 150 mM NaCl, 10% [v/v] glycerol, 10 mM dithiothreitol, 2% [w/v] polyvinyl-polypyrrolidone, 0.5% [v/v] Triton X-100, and 1×protease inhibitor cocktail) followed by centrifugation at $12,000 \times g$ and 4 °C for 30 min. The supernatant was incubated with 25 µl anti-FLAG (M2; Sigma-Aldrich) agarose beads or anti-GFP Trap-A beads (Chromotek) at 4 °C for 60 min. The beads were washed six times with IP buffer (25 mM Tris–HCl [pH 7.5], 1 mM EDTA, 150 mM NaCl, 10% [v/v] glycerol, 1 mM dithiothreitol, 0.1% [v/v] Triton X-100) by centrifuging at $1000 \times g$ and 4 °C for 1 min. Protein samples were heated at 95 °C for 5 min and separated by SDS-PAGE. The resulting blots were incubated with anti-FLAG-HRP antibodies (Sigma-Aldrich, Cat. # A8592; clone M2; 1:10,000) or anti-HA (Abcam, Cat. # ab18181; clone C5, 1:20,000), anti-YFP (Sigma-Aldrich Cat. # SAB4301138; 1:10,000), anti-GST (Sigma-Aldrich, Cat. # SAB1305539; clone 9AT106; 1:5000), anti-Myc (Sangon Biotech, Cat. #D191042; 1:5000); anti-RFP (made in this study, 1:5000), anti-NSm (made in this study, 1:5000), anti-Sw-5b (made in this study, 1:1000) at room temperature for 1.5 h or overnight at 4 °C, followed by HRP-conjugated goat-anti rabbit (Sigma-Aldrich, Cat # A0545, 1:10,000)/mouse (Sigma-Aldrich Cat # A4416; 1:10,000).

The blots were detected by the ECL Substrate Kit (Thermo Scientific, Hudson, NH, USA). For estimation of protein loading, the blots were stained with *Ponceau S*. Protein accumulation level was quantified by ImageJ. All experiments were repeated at least three times with similar results.

## Yeast two-hybrid assay

Yeast two-hybrid screen was performed as described (Zhu et al, 2014). Briefly, pGBKT7-SD or pGBKT7-NB was transformed into the yeast strain Y2H Gold, and mated with Y187 carrying cDNA library of *N. Benthamiana* in pGADT7 (generated in Clonetech) for 24 h at 30 °C on a 45 rpm shaker. The fresh yeast diploid cells generated by mating were plated on selective triple dropout media to detect protein-protein interactions. Positive clones were isolated to obtain plasmids for sequencing and the bait and prey were retransformed into the yeast strain Y2H Gold for the verification of interaction. The yeast transformed with BK and AD derivative constructs were plated on both synthetic Leu and Trp double dropout media (-TL) and synthetic Leu, Trp, His, and Ade quadruple dropout media (-TLHA).

## Split-luciferase complementation assay

Sw-5b SD, NB, SBP1, and SBP1$^{RM}$ were cloned into the pCambia1300-n/cLUC vector, respectively. Agrobacteria carrying the nLuc and cLuc constructs were mixed in a 1:1 proportion, and adjusted to $OD_{600} = 0.25$ in the induction medium. The agroinfiltrated *N. benthamiana* leaves were treated with 0.2 mg/mL D-luciferin (Yeasen, Shanghai, China) at 22 h post infiltration and incubated for 15 min. For competitive split-LUC assay, an increasing amounts of TSWV NSm or TZSV NSm were added to the Agrobacteria mixture of cLUC-SD + SBP1$^{RM}$-nLUC or cLUC-NB-LRR + SBP1$^{RM}$-nLUC. The leaves were treated with D-luciferin at 36 h post infiltration. Luminescence was detected by a low-light cooled charge-coupled device imaging apparatus (Vilber, FUSION FX7 IR SPECTRA). All experiments were repeated at least three times with similar results.

## Generation of knock out *N. benthamiana* plants by CRISPR/Cas9

The *sbp1* CRISPR/Cas9-edited mutant lines were generated as described (Ma et al, 2015; Fu et al, 2018). Briefly, the fragment containing single sgRNA was cloned into pGBK01/Cas9 vector (Fu et al, 2018) to generate pGBK01/Cas9-SBP1. Single guide RNA (sgRNA) was designed to target the open reading frame (ORF) of *SBP1* using CRISPR-P tool (http://cbi.hzau.edu.cn/crispr/) (Lei et al, 2014). The sgRNA sequences are listed in Appendix Table S1. The CRISPR constructs were transformed into wild-type *N. benthamiana* leaf discs by Agrobacterium-mediated transformation. The CRISPR target sites in the resulting transgenic plants were amplified using PCR and sequenced. The progeny of homozygous *sbp1-1* and *sbp1-2* line was used as experimental materials.

## Trichloroacetic acid-acetone protein precipitation

Trichloroacetic acid (TCA)-Acetone protein precipitation was performed as described (Méchin et al, 2007; Niu et al, 2018).

Briefly, total protein was extracted from 2.0 g *Solanum lycopersicum* leaves in 2 mL of extraction buffer (25 mM Tris–HCl [pH 7.5], 1 mM EDTA, 150 mM NaCl, 10% [v/v] glycerol, 10 mM dithiothreitol, 2% [w/v] polyvinylpolypyrrolidone, 0.5% [v/v] Triton X-100, 0.02% Na deoxycholate [DOC], and 1×protease inhibitor cocktail) followed by 30 min incubation at 4 °C. 100% trichloroacetic acid (TCA) was added to the extract to get a final concentration of 10% TCA. The mixture was incubated at 4 °C overnight, followed by centrifugation at 15,000 × g and 4 °C for 30 min. The supernatant was discarded and 1 mL ice-cold acetone was added to resuspend the pellet. The tube was kept on ice at least 15 min followed by centrifugation at 15,000 × g and 4 °C for 10 min. The supernatant was discarded and the pellet was dissolved in a minimal volume of sample buffer followed by immunoblotting.

## VIGS assay

VIGS assay were performed in tomato cultivar 1760 carrying *Sw-5b* resistance gene. A 300 bp fragment from cDNA of *SlSBP1* was selected based on SNG-VIGS tool (https://vigs.solgenomics.net/), and cloned into pTRV2 vector (Liu et al, 2002). Agrobacterium carrying TRV1 and Agrobacterium carrying TRV2-SlSu, TRV2-SlSBP1, or TRV2-GUS construct were mixed in 1:1 proportion and infiltrated into leaves of 4-week-old plants. When plant treated with TRV-SlSu showed photobleaching phenotype in the upper leaves, the corresponding upper leaves of TRV-SlSBP1 and TRV-GUS treated plants were used for isolation of total RNA for qRT-PCR or extraction of total protein for TCA-Acetone precipitation followed by western blot analysis.

## Real-time qRT-PCR

qRT-PCR analysis was performed as previously described (Huang et al, 2020). Total RNA from *N. benthamiana* leaves expressing different constructs or from plants pretreated with TRV constructs was extracted using RNA-easy isolation reagent (Vazyme; catalog no. R701-01). The cDNA was prepared from 1 μg RNA using oligo d(T) primer and HiScript® III 1st Strand cDNA Synthesis Kit (Vazyme; catalog no. R312-01). qPCR was performed using ChamQ Universal SYBR qPCR Master Mix (Vazyme; catalog no. Q711-02) in the CFX96 Touch Real-Time PCR Detection System (Bio-Rad).

## Protein expression and purification

Constructs pGEX-2TK-Sw-5b, pGEX-2TK-Sw-5b$^{D857V}$, pGEX-2TK-SD, pGEX-2TK-NB-LRR, pET28a-SBP1-HA-His, and pET28a-SBP1$^{RM}$-HA-His were individually transformed into *E. coli* Rosetta (DE3) strain. To express the recombinant protein, about 10 mL overnight culture were transferred to 1 L LB and incubated at 37 °C till OD$_{600}$ reached 0.6–0.8. Protein expression was induced with 0.1 mM IPTG (isopropyl b-D-thiogalactopyranoside) for 16 h at 20 °C. Purification of GST-fused protein and His-fused protein were performed as described previously (Li et al, 2019).

## In vitro and in vivo ubiquitination assay

In vitro ubiquitination assay of SBP1 was performed as described with slight modification (Liu et al, 2022). UBA1-6×His (E1), UBC8-6×His (E2), 6×His-FLAG-Ub, SBP1-HA-6×His, SBP1$^{RM}$-HA-6×His, GST-Sw-5b, GST-Sw-5b$^{D857V}$, GST-SD and GST-NB-LRR was expressed and purified from *E. coli*. The purified proteins were dialyzed in a lysis buffer (80 mM Tris–HCl [pH 7.5], 50 mM NaCl) at room temperature for 1 h. For in vitro ubiquitination assays, the purified UBA1-6×His (E1), UBC8-6×His (E2), 6×His-FLAG-Ub, SBP1-HA-6×His, GST-Sw-5b, GST-Sw-5b$^{D857V}$, GST-SD and GST-NB-LRR were mixed in different combinations in an ubiquitination buffer (80 mM Tris–HCl [pH 7.5], 25 mM MgCl$_2$, 12 mM ATP; 30 μL per sample) and incubated, for 2 h at 26 °C. Alternatively, the LOF SBP1$^{RM}$ mutant was used in place of WT SBP1. Reactions without 6×His-FLAG-Ub or SBP1-HA-6×His were used as control. The reaction was ended by adding of 1×SDS loading buffer followed by 5 min incubation at 95 °C.

For in vivo ubiquitination assay, YFP-Sw-5b and FLAG-Ub were co-expressed in WT or *sbp1-1* plant in the presence of 25 μM MG132. All the ubiquitinated proteins were immunoprecipitated with FLAG-specific antibodies, the ubiquitination of YFP-Sw-5b was detected by immunoblotting the IP ubiquitinated proteins using YFP-specific antibodies.

To evaluate the effect of NSm on the ubiquitination of Sw-5b mediated by SBP1 in vitro, TSWV NSm-Myc-His or TZSV NSm-Myc-His was added to the ubiquitination assay mixture of Sw-5b and incubated at 26 °C for 2 h. The proteins were detected by immunoblots using GST, HA, FLAG, and Myc-specific antibodies.

To evaluate the effect of NSm on the ubiquitination of Sw-5b mediated by SBP1 in vivo, YFP-Sw-5b and FLAG-Ub were co-expressed with TSWV NSm-HA or TZSV NSm-HA for 24 h. The ubiquitinated proteins in plants were immunoprecipitated with FLAG-specific antibodies. The overall ubiquitination of endogenous proteins was detected by immunoblotting using FLAG-specific antibodies and the ubiquitination of YFP-Sw-5b were detected using YFP-specific antibodies.

## Mass spectrometry determination of ubiquitination sites

YFP-SD and YFP-NB-LRR were expressed in *N. benthamiana* leaves for 24 h and immunoprecipitated as previously described (Zhu et al, 2017). The proteins bound to the beads were released by boiling and separated by SDS-PAGE. The corresponding bands on gel were cut and trypsin-digested prior to LC–MS/MS analysis on an Orbitrap Exploris 480 (Thermo Fisher). The MS/MS spectra were analyzed with Proteome Discoverer software (Thermo Fisher Scientific, version 3.0).

## In vitro and in vivo time course degradation assay

For in vivo time-course protein degradation, YFP-Sw-5b, YFP-Sw-5b$^{D857V}$ or different Sw-5b mutants were expressed in WT or *sbp1-1* *N. benthamiana* plants leaves. The leaves were treated with 10 μg/mL cycloheximide (CHX) at 20 h post agroinfiltration (hpi) to block protein synthesis. Samples from different leaves were taken at 0–3 h after CHX treatment and the protein level on the blots were detected using YFP-specific antibodies. The assay were repeated at least three times with similar results.

For in vitro time course protein degradation assay, GST-Sw-5b and GST-Sw-5b$^{D857V}$ was purified from *E. coli* and incubated with total protein extracts from WT *N. benthamiana* at room temperature for 0–3 h. Protein level on the blot was detected using GST-specific antibodies.

## Competitive GST-pull down assay

For competitive pull-down assays, GST or GST-Sw-5b was incubated with 30 μL glutathione-agarose beads for 3 h at 4 °C. The beads were then washed four times with IP buffer and incubated with SBP1-HA-His. Subsequently, an increasing amount of TSWV NSm-Myc-His or TZSV NSm-Myc-His protein (1, 5, 10 μg) were added to the mixture and incubated at 4 °C for 3 h. Following four washes, the proteins bound to the beads were separated by SDS-PAGE and detected using GST, HA, and Myc-specific antibodies.

## Cell death assay and ion leakage measurement

Development of cell death in leaves was monitored through daily observation in the infiltrated leaves and ion leakage was measured as previously described (Zhu et al, 2017). Briefly, five leaf discs of 5 mm diameter harvested from the infiltrated leaves at 48 hpi and floated on 10 mL of double distilled water at room temperature for 3 h. After incubation, the solution without leaf discs was measured with a conductivity meter (Seven Excellence, METTLER TOLEDO) and referred to as value A. The leaf discs were then returned to the solution and treated at 95 °C for 25 min. Then, the solution was measured and referred to as value B. The conductivity was expressed as percent ion leakage: (value A/value B) × 100%. Statistical analyses were performed using GraphPad Prism 8.0.

## Quantification and statistical analysis

All protein band shows in western blots were quantified by ImageJ (https://imagej.nih.gov/ij/index.html). Statistical significance analysis was examined by unpaired two-tailed Student's t-test or by one-way ANOVA with Tukey's test using GraphPad software (Chen et al, 2023; Huang et al, 2024).

# Data availability

All supporting data of this study are available from the article and Supplementary Information files, or from the corresponding author upon request. Source data are provided with this paper.

The source data of this paper are collected in the following database record: biostudies:S-SCDT-10_1038-S44318-024-00174-6.

# Peer review information

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

## Acknowledgements

We thank Prof. Sophien Komoun (The Sainsbury Laboratory, Norwich, UK) for kindly providing the plasmids of R8 and Rpi-blb2, Prof. Dongping Lv (Shanghai Jiaotong University) the plasmids of UBA1-His (E1) and UBC8-His (E2), and Prof. Shunping Yan (Huazhong Agricultural University) for the plasmids of His-FLAG-Ub. This work was supported by grants from the National Natural Science Foundation of China (32220103008, 31925032, and 32272488), the National Key Research and Development Program of China (2022YFF1001500 and 2022YFD1401200), the Funds from the Independent Innovation of Agricultural Science and Technology of Jiangsu Province [Grant No. CX (22) 2039], the Jiangsu Key Technology R & D Program and International Science and Technology Cooperation Project (BZ2023030), the key projects of YNTC (2021530000241015), the Guidance Foundation of the Sanya Institute of Nanjing Agricultural University (NAUSY-MS19), and the Key Science and Technology Program of Hainan Province (ZDKJ2021007) to XT.

## Author contributions

**Chunli Wang**: Data curation; Investigation; Writing—original draft; Writing—review and editing. **Min Zhu**: Writing—review and editing. **Hao Hong**: Software; Writing—review and editing. **Jia Li**: Writing—review and editing. **Chongkun Zuo**: Data curation. **Yu Zhang**: Data curation. **Yajie Shi**: Data curation. **Suyu Liu**: Data curation; Investigation. **Haohua Yu**: Data curation. **Yuling Yan**: Data curation. **Jing Chen**: Data curation. **Lingna Shangguan**: Data curation. **Aiping Zhi**: Data curation. **Rongzhen Chen**: Data curation. **Karen Thulasi Devendrakumar**: Writing—review and editing. **Xiaorong Tao**: Supervision; Writing—review and editing.

Source data underlying figure panels in this paper may have individual authorship assigned. Where available, figure panel/source data authorship is listed in the following database record: biostudies:S-SCDT-10_1038-S44318-024-00174-6.

## Disclosure and competing interests statement

The authors declare no competing interests.

# Expanded View Figures

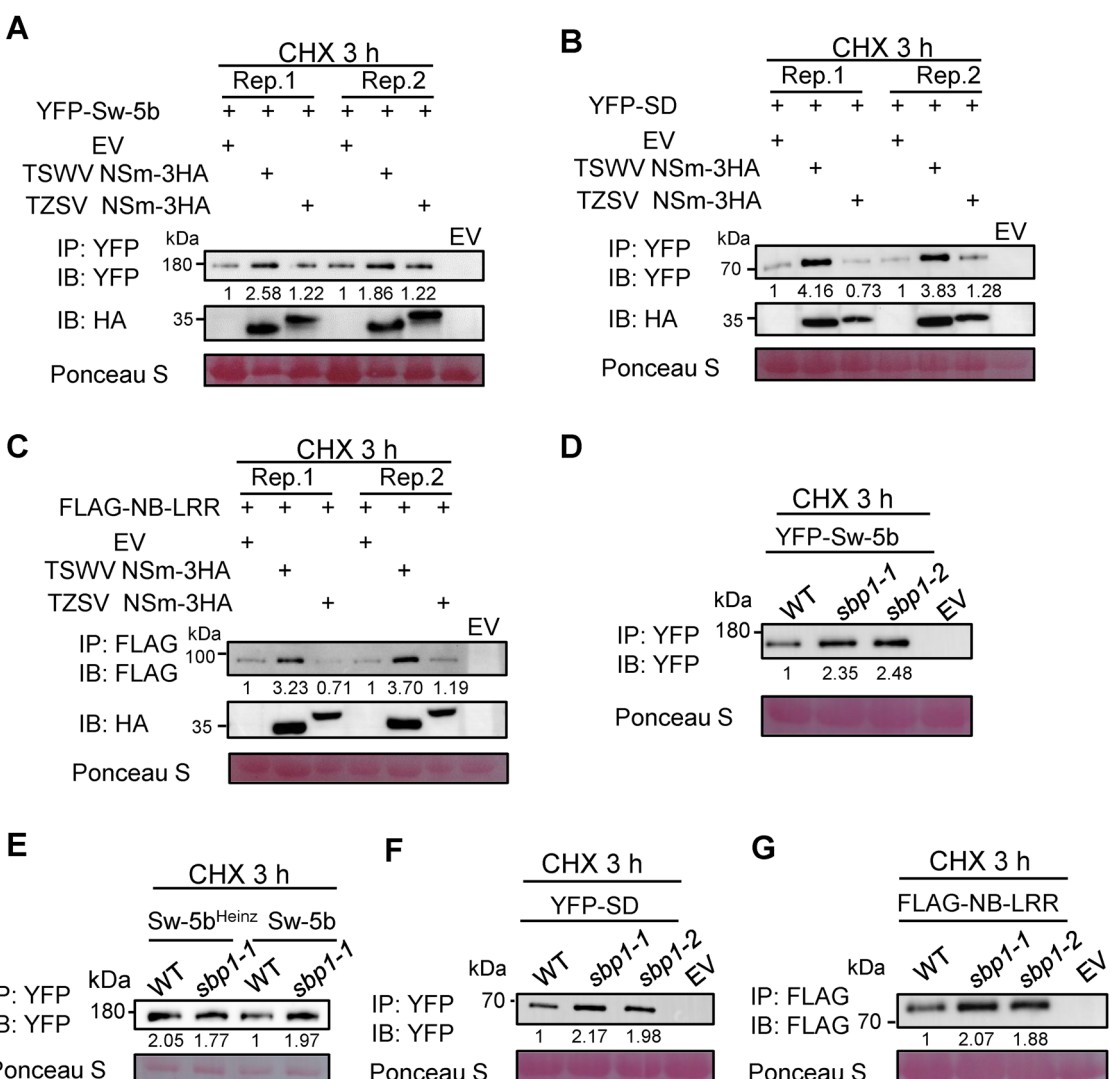

**Figure EV1. The TSWV NSm protects Sw-5b NLR from degradation by the plant 26S proteasome.**

(A–C) The accumulation of Sw-5b, SD and NB-LRR in the presence of TSWV NSm or TZSV NSm in *N. benthamiana* leaves. YFP-Sw-5b (A), YFP-SD (B) or FLAG-NB-LRR (C) was co-expressed with the pCambia2300 empty vector (EV), TSWV NSm, or TZSV NSm for 22 h. Samples were treated with 10 μg/mL cycloheximide (CHX) at 19 hpi to block protein synthesis. Protein accumulation was detected at 3 h post CHX treatment by immunoblot using YFP, FLAG, and HA-specific antibodies. (D–G) Protein accumulation analysis of YFP-Sw-5b (D), YFP-Sw-5b^Heinz (E), YFP-SD (F), and FLAG-NB-LRR (G) in wild-type (WT) and *SBP1* knockout *N. benthamiana* plants. YFP-Sw-5b, YFP-Sw-5b^Heinz, YFP-SD and FLAG-NB-LRR were expressed in WT, *sbp1-1* or *sbp1-2* mutant *N. benthamiana* leaves and treated with 10 μg/mL CHX at 21 hpi. Total protein was extracted from the samples 24 hpi and analyzed for protein accumulation by immunoblot using YFP and FLAG-specific antibody. Protein accumulation levels were quantified by ImageJ software. Source data are provided as a Source Data file. All experiments were repeated at least three times with similar results. Source data are available online for this figure.

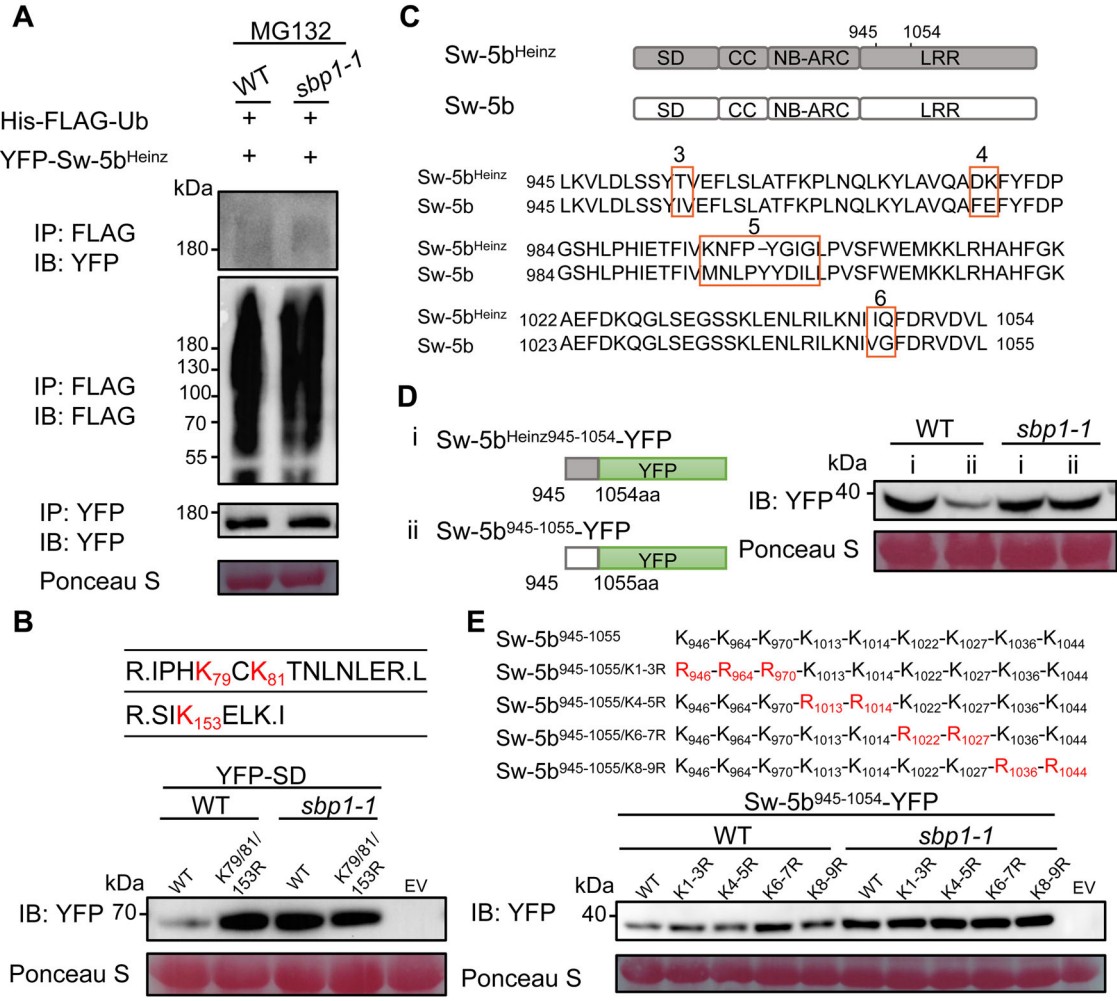

**Figure EV2. Determination of key amino acids within Sw-5b SD and LRR3-6 that were ubiquitinated by SBP1.**

(**A**) In vivo ubiquitination assay to analyze the ubiquitination of Sw-5b$^{Heinz}$ in WT and *SBP1* knockout *N. benthamiana* plants. YFP-Sw-5b$^{Heinz}$ was co-expressed with FLAG-Ub in WT or *sbp1-1* mutant *N. benthamiana* leaves, 25 μM MG132 was infiltrated into the leaves at 16 hpi. The ubiquitination of Sw-5b$^{Heinz}$ was detected at 24 hpi by immunoprecipitation using anti-FLAG beads followed by detection with YFP-specific antibodies. The overall ubiquitinated proteins were detected using FLAG-specific antibodies. (**B**) Determination of amino acids within Sw-5b SD that was ubiquitinated by SBP1. The ubiquitinated lysine residues K79, K81, and K153 identified by LC–MS/MS are shown in red (upper panel). The protein accumulation of wild-type SD and SD$^{K79/81/153R}$ mutant was analyzed in wild-type and *sbp1-1* mutant *N. benthamiana* plants (lower panel). (**C–E**) Determination of amino acids within Sw-5b LRR3-6 that was ubiquitinated by SBP1. The schematic diagram of Sw-5b and Sw-5b$^{Heinz}$ homolog showing the SD, CC, NB-ARC and LRR domains is shown in the top panel (**C**). The polymorphic, 945–1054 amino acid (aa) region encompassing polymorphic sites 3–6 of the LRR domain (LRR3-6) of the Sw-5b and the Sw-5b$^{Heinz}$ is shown in the bottom panel (**C**). The key amino acids for NSm recognition is shown in the red boxed region. The 945–1054 aa region of Sw-5b$^{Heinz}$ or the 945–1055 aa region of Sw-5b encompassing LRR3-6 was fused to the N-terminus of YFP. The accumulation levels of Sw-5b$^{Heinz945-1054}$-YFP and Sw-5b$^{945-1055}$-YFP were analyzed in WT or *sbp1-1* mutant leaves at 24 hpi by immunoblot using YFP-specific antibodies (**D**). All lysine sites in Sw-5b LRR3-6 were analyzed by mutating K946-K964-K970, K1013-K1014, K1022-K1027, K1036-K1044 to R in four mutants. Arginine used to replace lysine sites in Sw-5b$^{945-1055}$ mutants are shown in red. The protein accumulation levels of Sw-5b$^{945-1055}$-YFP and its mutants were analyzed in WT and *sbp1-1* mutant *N. benthamiana* plants using YFP-specific antibodies (**E**). Source data are provided as a Source data file. All experiments were repeated at least three times with similar results. Source data are available online for this figure.

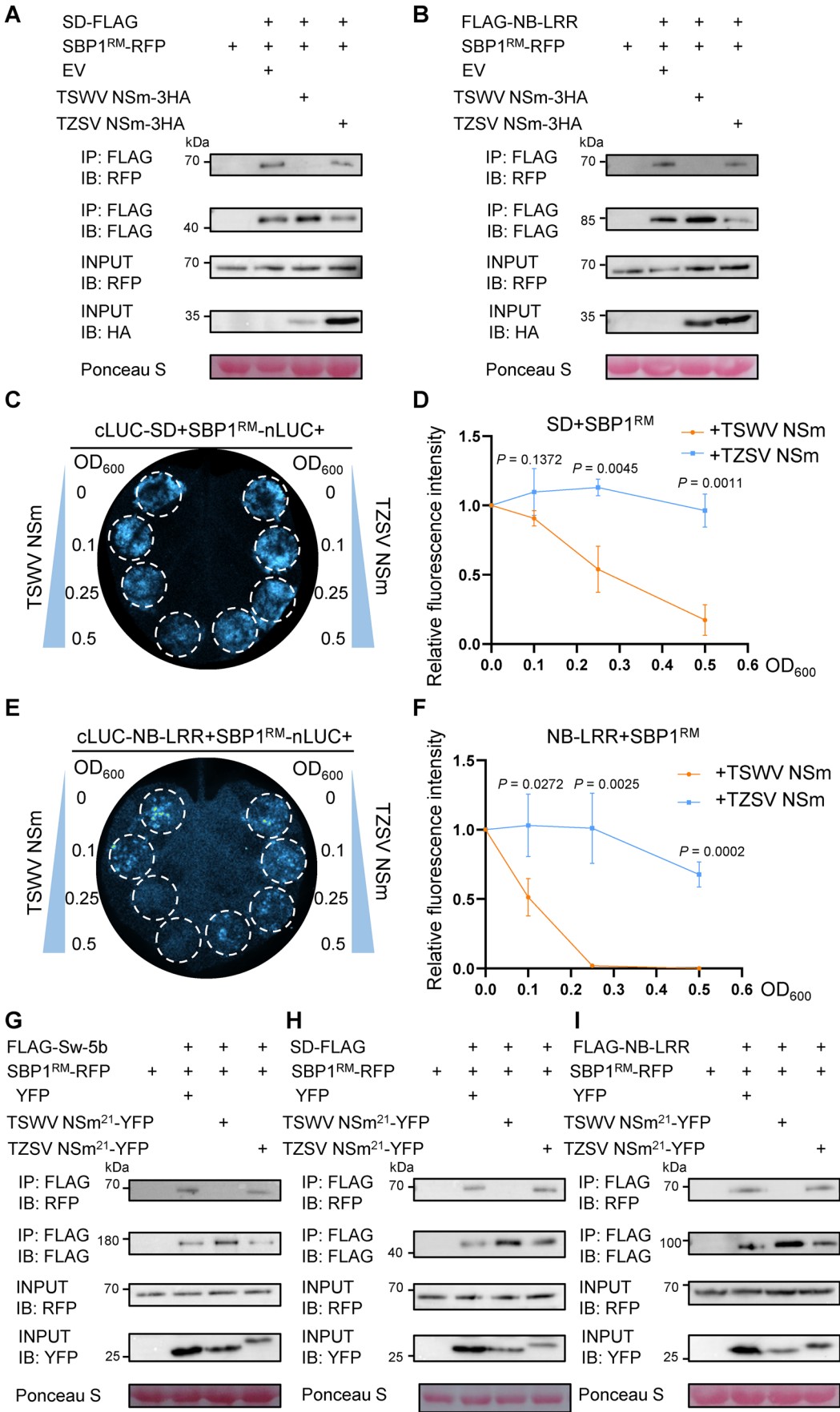

◀ **Figure EV3.  Tospoviral NSm and conserved NSm[21] peptide region interferes with the interaction between Sw-5b and SBP1.**

(**A, B**) TSWV NSm interferes with the interactions of Sw-5b SD and NB-LRR with SBP1. SD-FLAG (**A**) or FLAG-NB-LRR (**B**) was immunoprecipitated with SBP1[RM]-RFP in the presence of EV, TSWV NSm, or TZSV NSm, with SBP1[RM]-RFP alone as a control. The co-IP assay was carried out at 21 hpi and immunoblots were probed using YFP, FLAG, RFP, and HA-specific antibodies. (**C, E**) Split-luciferase complementation assay showing the inhibitory effects of TSWV NSm on the interactions of Sw-5b SD (**C**) and NB-LRR (**E**) with SBP1. The cLUC-SD or cLUC-NB-LRR was co-expressed with SBP1[RM]-nLUC in the presence of increasing amounts of Agrobacteria carrying TSWV NSm or TZSV NSm. Luciferase activity was detected at 36 hpi. (**D, F**) The fluorescence intensity of the interaction between SD (**D**) and NB-LRR (**F**) with SBP1[RM] from (**C**) and (**E**) was quantified using ImageJ. Data are presented as means ± SD ($n = 3$ biologically independent samples). Data were analyzed by two-sided Student's $t$-test. The exact $P$ values are indicated in the graphs. (**G–I**) TSWV NSm[21]-YFP interferes with the interaction of Sw-5b, SD, and NB-LRR with SBP1. YFP-Sw-5b (**G**), SD-FLAG (**H**), or FLAG-NB-LRR (**I**) was immunoprecipitated with SBP1[RM]-RFP in the presence of EV, TSWV NSm[21]-YFP, or TZSV NSm[21]-YFP. The co-IP assay was carried out at 21 hpi and immunoblots were probed using YFP, RFP, and FLAG-specific antibodies. The size of protein was shown on the left. IB, immunoblot with specific antibody; IP, immunoprecipitation with specific antibody. Ponceau S staining was used to show the amount of protein loaded. Experiments were repeated at least three times with similar results. Source data are provided as a Source data file. Source data are available online for this figure.

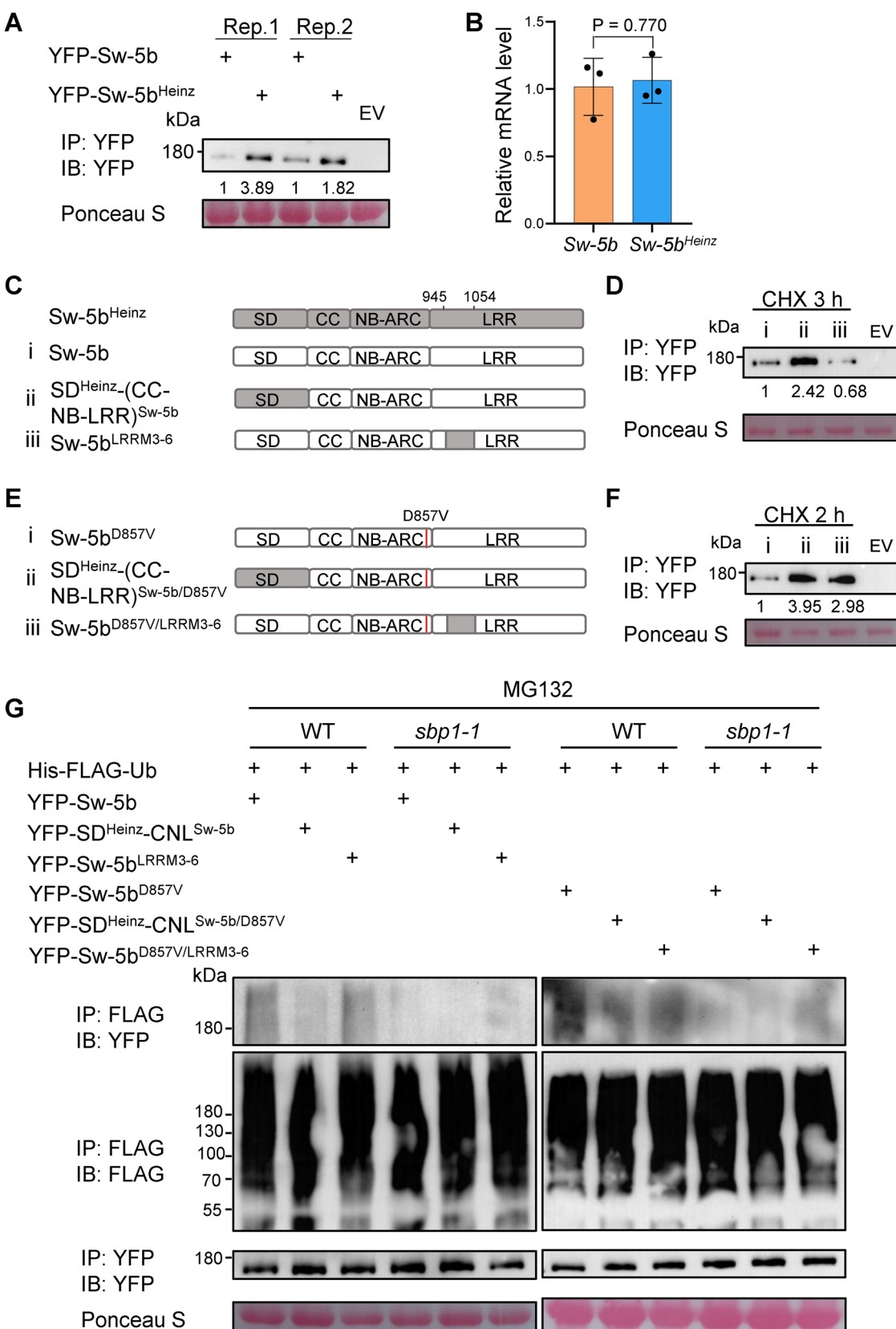

**Figure EV4. Protein accumulation and ubiquitination analysis of chimeric Sw-5b variants.**

(A) The accumulation of Sw-5b and Sw-5b[Heinz] in *N. benthamiana* leaves. YFP-Sw-5b was expressed in one-half leaf of *N. benthamiana* plant and YFP-Sw-5b[Heinz] expressed in another half of the same leaf. Protein accumulation was detected by immunoblot at 24 hpi using YFP-specific antibody. (B) The relative transcript expression levels of YFP-Sw-5b and YFP-Sw-5b[Heinz]. Total RNA was extracted at 24 hpi for qRT-PCR analysis. Data are presented as means ± SD ($n = 3$ biologically independent samples). Data were analyzed by two-sided Student's *t*-test. The exact *P* values are indicated in the graph. (C) The schematic diagram showing the architecture of the chimeric proteins used to assay the role of SD and LRR (945–1054 aa) in maintaining the homeostasis of the inactive Sw-5b. (D) The accumulation of YFP-Sw-5b, YFP-SD[Heinz]-(CC-NB-LRR)[Sw-5b], and YFP-Sw-5b[LRRM3-6] in *N. benthamiana* leaves. The leaves were treated with 10 µg/mL CHX at 21 hpi and protein levels were analyzed by immunoblot 3 h post CHX treatment. (E) The schematic diagram showing the architecture of the chimeric proteins used to assay the role of SD and LRR (945–1054 aa) in maintaining the homeostasis of the constitutively active Sw-5b (Sw-5b[D857V]). (F) The accumulation of YFP-Sw-5b[D857V], YFP-SD[Heinz]-(CC-NB-LRR)[Sw-5bD857V], and YFP-Sw-5b[D857V/LRRM3-6] in *N. benthamiana* leaves at 21 hpi. The leaves were treated with 10 µg/mL CHX at 19 hpi and protein levels were analyzed by immunoblot 3 h post CHX treatment. In (A), (D), and (F), IB, immunoblot with specific antibody; IP, immunoprecipitation with specific antibody. Ponceau S staining was used to show the amount of protein loaded. Protein accumulation level was quantified by ImageJ software. (G) In vivo ubiquitination assay of Sw-5b and chimeric Sw-5b variants in WT and *SBP1* knockout *N. benthamiana* plants. YFP-Sw-5b or chimeric Sw-5b variants was co-expressed with FLAG-Ub in leaves of WT or *sbp1-1* mutant *N. benthamiana* leaves, 25 µM MG132 was infiltrated at 16 hpi. The ubiquitination of Sw-5b was detected at 24 hpi by immunoprecipitation using anti-FLAG beads followed by detection with YFP-specific antibodies. The overall ubiquitinated proteins were detected using FLAG-specific antibodies. Experiments were repeated at least three times with similar results. Source data are provided as a Source data file. Source data are available online for this figure.

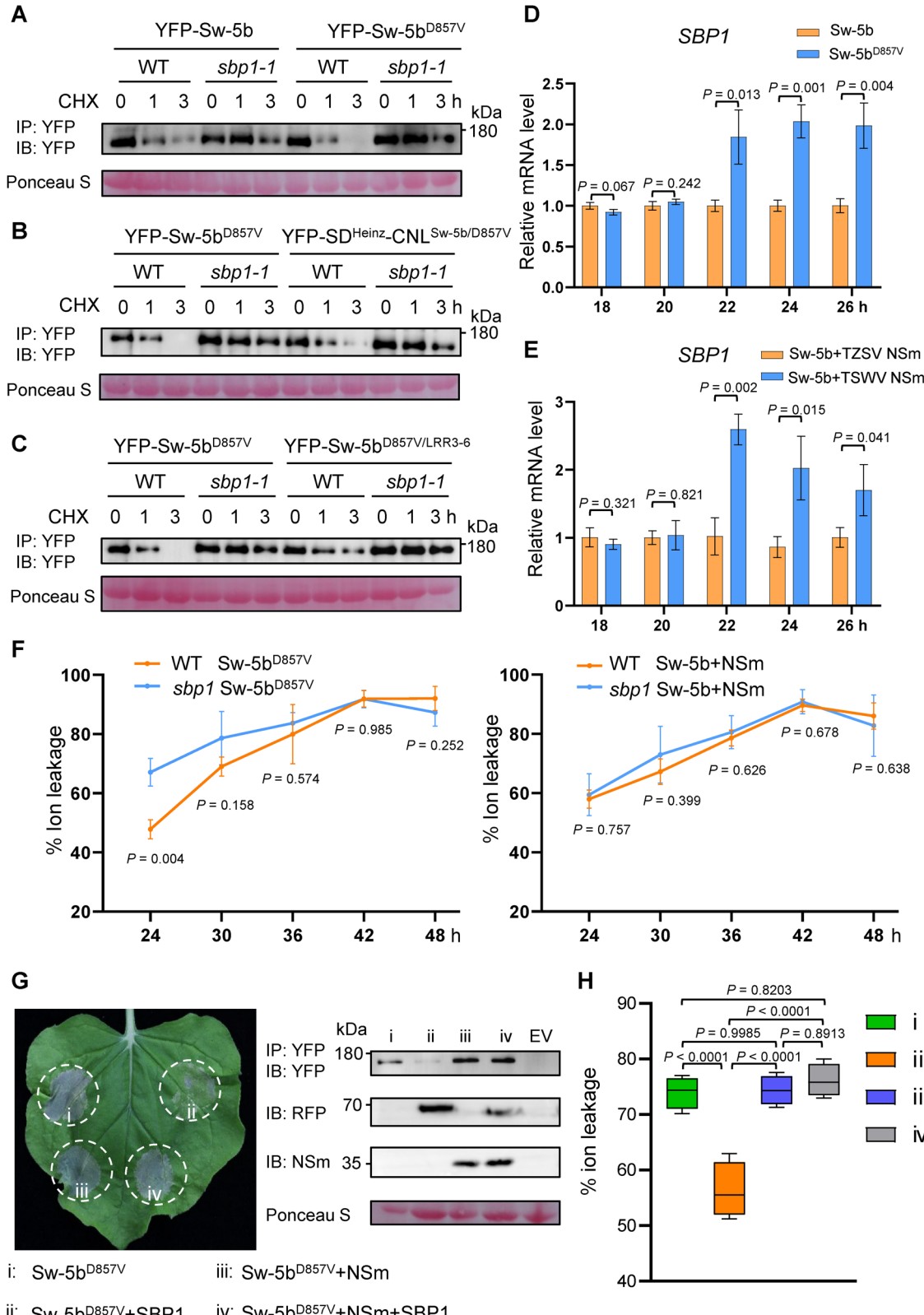

**G**

i: Sw-5b^D857V

ii: Sw-5b^D857V+SBP1

iii: Sw-5b^D857V+NSm

iv: Sw-5b^D857V+NSm+SBP1

**Figure EV5.  Homeostasis of active state Sw-5b is regulated by the E3 ligase SBP1 and the TSWV effector NSm.**

(A) Protein turnover rate of Sw-5b and Sw-5b$^{D857V}$ in leaves of WT or *sbp1-1* mutant *N. benthamiana* plants. YFP-Sw-5b or YFP-Sw-5b$^{D857V}$ was expressed in *N. benthamiana* leaves and treated with 10 μg/mL CHX at 20 hpi to block protein synthesis. Samples were taken at 0–3 h post CHX treatment. The samples were analyzed by immunoblot using YFP-specific antibodies. (B) Protein turnover rate of Sw-5b$^{D857V}$ and SD$^{Heinz}$-CNL$^{Sw-5b/D857V}$ in leaves of WT or *sbp1-1* mutant *N. benthamiana* plants. The protein accumulation was examined as described in (A). (C) Protein turnover rate of Sw-5b$^{D857V}$ and Sw-5b$^{D857V/LRR3-6}$ in leaves of WT or *sbp1-1* of *N. benthamiana* plants. The protein accumulation was examined as described in (A). (D) The time course analysis of relative mRNA expression level of *SBP1* in *N. benthamiana* leaves expressing Sw-5b or Sw-5b$^{D857V}$. (E) The time course analysis of relative mRNA expression levels of *SBP1* in *N. benthamiana* leaves co-expressing Sw-5b and elicitor TSWV NSm or non-elicitor TZSV NSm. (F) Ion leakage analysis of leaves of WT and *sbp1-1 N. benthamiana* plants co-expressing Sw-5b and NSm or expressing Sw-5b$^{D857V}$ at 6-h intervals from 24 to 48 h post agroinfiltration. NSm here refers to the elicitor TSWV NSm. For (D), (E), and (F), data are presented as means ± SD ($n = 3$ biologically independent samples). The exact *P* values are indicated in the graphs. Data were analyzed by two-sided Student's *t*-test. (G, H) HR cell death and ion leakage analysis of YFP-Sw-5b$^{D857V}$ and YFP-Sw-5b$^{D857V}$ + NSm in the absence or the presence of SBP1-RFP in *N. benthamiana* leaves. NSm refers to the elicitor TSWV NSm. HR phenotype in the infiltrated leaves was photographed at 4 days post inoculation (G). Protein accumulation in (G) was detected by immunoblotting using YFP, RFP, and NSm-specific antibodies. The ion leakage was measured at 2 dpi after agroinfiltration and shown in (H). Data are shown as the box plots with the interquartile range as the upper and lower confines, minima and maxima as whiskers, and the median as a solid line ($n = 4$); The exact *P* values are indicated in the graph (one-way ANOVA). Source data are provided as a Source data file. All experiments were repeated at least three times with similar results. Source data are available online for this figure.

