## [Peer Review File · The EMBO Journal]

A viral effector blocks the turnover of a plant NLR receptor to trigger a robust immune response

Chunli Wang, Min Zhu, Hao Hong, Jia Li, Chongkun Zuo, Yu Zhang, Yajie Shi, Suyu Liu, Haohua Yu, Yuling Yan, Jing Chen, Lingna Shanguan, Aiping Zhi, Rongzhen Chen, Karen Thulasi Devendrakumar, and Xiaorong Tao

Corresponding author(s): Xiaorong Tao (taoxiaorong@njau.edu.cn)

Review Timeline:

Submission Date:	24th Dec 23
Editorial Decision:	14th Feb 24
Revision Received:	2nd May 24
Editorial Decision:	13th Jun 24
Revision Received:	16th Jun 24
Accepted:	24th Jun 24

Editor: William Teale

Transaction Report:

Dear Dr. Tao,

Thank you again for the submission of your manuscript entitled "E3-mediated turnover of OFF/ON states of plant NLR converted by pathogen effector to robust immunity" (EMBOJ-2024-116497) and for your patience during the review process. We have now received the reports from the referees, which I copy below.

As you can see from their comments, while referees 2 and 3 provide lists of recommended controls, all of them point out the potential significance of your work.

Based on this overall interest expressed in the reports, I would like to invite you to address the comments of all referees in a revised version of the manuscript. I should add that it is The EMBO Journal policy to allow only a single major round of revision and that it is therefore important to resolve the main concerns at this stage. I believe the concerns of the referees are reasonable and addressable, but please contact me if you have any questions, need further input on the referee comments or if you anticipate any problems in addressing any of their points. I recommend we discuss the experimental plan for the revised version of the manuscript over Zoom when you have had time for a careful look. Please contact me with some convenient time-slots. Please also follow the instructions below when preparing your manuscript for resubmission.

I would also like to point out that as a matter of policy, competing manuscripts published during this period will not be taken into consideration in our assessment of the novelty presented by your study ("scooping" protection). We have extended this 'scooping protection policy' beyond the usual 3 month revision timeline to cover the period required for a full revision to address the essential experimental issues. Please contact me if you see a paper with related content published elsewhere to discuss the appropriate course of action.

Again, please contact me at any time during revision if you need any help or have further questions.

Thank you very much again for the opportunity to consider your work for publication. I look forward to your revision.

Best regards,

William

William Teale, Ph.D.
Editor
The EMBO Journal

When submitting your revised manuscript, please carefully review the instructions below and include the following items:

- 1) a .docx formatted version of the manuscript text (including legends for main figures, EV figures and tables). Please make sure that the changes are highlighted to be clearly visible.
- 2) individual production quality figure files as .eps, .tif, .jpg (one file per figure).
- 3) a .docx formatted letter INCLUDING the reviewers' reports and your detailed point-by-point response to their comments. As part of the EMBO Press transparent editorial process, the point-by-point response is part of the Review Process File (RPF), which will be published alongside your paper.
- 4) a complete author checklist, which you can download from our author guidelines ([https://wol-prod-cdn.literatumonline.com/pb-assets/embo-site/Author Checklist%20-%20EMBO%20J-1561436015657.xlsx](https://wol-prod-cdn.literatumonline.com/pb-assets/embo-site/Author%20Checklist%20-%20EMBO%20J-1561436015657.xlsx)). Please insert information in the checklist that is also reflected in the manuscript. The completed author checklist will also be part of the RPF.
- 5) Please note that all corresponding authors are required to supply an ORCID ID for their name upon submission of a revised manuscript.
- 6) We require a 'Data Availability' section after the Materials and Methods. Before submitting your revision, primary datasets produced in this study need to be deposited in an appropriate public database, and the accession numbers and database listed under 'Data Availability'. Please remember to provide a reviewer password if the datasets are not yet public (see

<https://www.embopress.org/page/journal/14602075/authorguide#datadeposition>). If no data deposition in external databases is needed for this paper, please then state in this section: This study includes no data deposited in external repositories. Note that the Data Availability Section is restricted to new primary data that are part of this study.

Note - All links should resolve to a page where the data can be accessed.

8) For data quantification: please specify the name of the statistical test used to generate error bars and P values, the number (n) of independent experiments (specify technical or biological replicates) underlying each data point and the test used to calculate p-values in each figure legend. The figure legends should contain a basic description of n, P and the test applied. Graphs must include a description of the bars and the error bars (s.d., s.e.m.).

9) We would also encourage you to include the source data for figure panels that show essential data. Numerical data can be provided as individual .xls or .csv files (including a tab describing the data). For 'blots' or microscopy, uncropped images should be submitted (using a zip archive or a single pdf per main figure if multiple images need to be supplied for one panel). Additional information on source data and instruction on how to label the files are available at .

10) We replaced Supplementary Information with Expanded View (EV) Figures and Tables that are collapsible/expandable online (see examples in <https://www.embopress.org/doi/10.15252/embj.201695874>). A maximum of 5 EV Figures can be typeset. EV Figures should be cited as 'Figure EV1, Figure EV2" etc. in the text and their respective legends should be included in the main text after the legends of regular figures.

12) Our journal encourages inclusion of *data citations in the reference list* to directly cite datasets that were re-used and obtained from public databases. Data citations in the article text are distinct from normal bibliographical citations and should directly link to the database records from which the data can be accessed. In the main text, data citations are formatted as follows: "Data ref: Smith et al, 2001" or "Data ref: NCBI Sequence Read Archive PRJNA342805, 2017". In the Reference list, data citations must be labeled with "[DATASET]". A data reference must provide the database name, accession number/identifiers and a resolvable link to the landing page from which the data can be accessed at the end of the reference. Further instructions are available at .

We realize that it is difficult to revise to a specific deadline. In the interest of protecting the conceptual advance provided by the work, we recommend a revision within 3 months (14th May 2024). Please discuss the revision progress ahead of this time with the editor if you require more time to complete the revisions. Use the link below to submit your revision:

Referee #1:

Plant NLR immune receptors are usually triggered, directly or indirectly, by pathogen effectors and in the absence of pathogen attack various transcriptional and post-transcriptional mechanisms (including E3 Ub ligase turnover) have been characterized which keep NLRs at low levels and inactive to reduce fitness trade-offs. This study investigates NLR homeostatic processes regulating the function of a tomato CC-domain NLR receptor, Sw-5b, building on previous published insights by this group on Sw-5b functional activation by a viral effector. This is an interesting question because, although it is known that multi-domain NLR proteins can shift from auto-inhibited to primed (intermediate) and fully active (immune triggering) states, the processes underlying NLR state equilibrium are not fully resolved. The work presented here goes significantly beyond earlier studies and is important for the field. It demonstrates in a comprehensive and convincing way how the on/off state of the Sw-5b SD-CC-NB-LRR protein is controlled by a Ub E3 ligase (SBP1). Two aspects make the analysis interesting and novel: (i) characterizing the genetic and molecular basis of SBP1 differential control of Sw-5b turnover as an inhibited (inactive) form or as a conformationally activated form, and (ii) the mode of viral effector interference with SBP1-mediated degradation of Sw-5b through direct effector binding to Sw-5b domains to conformationally activate the NLR and compete with SBP1-NLR degradation - which leads to 'released' effector-triggered immunity. The data provide solid validation of previous NLR on-off equilibrium models, but also a detailed mode of action, by which different NLR states are maintained in healthy and pathogen-activated plants. The authors use complementary yeast, plant (Nb and tomato), in vitro assays and suitable controls (eg. an autoactive Ws-5b variant and SW-5b effector recognition competent and non-competent forms/chimeras) to test their model. The sum of the experiments is a compelling demonstration of processes underlying a CC-NLR state equilibrium and immune function. The Results are presented clearly and the Discussion is balanced and generally informative.

I don't have specific queries - more of a comment which authors might consider expanding on a bit more in Discussion. Although they don't have access to structurally resolved Sw-5b states, the fact that individual SD and NB-LRR domains are also targeted by SBP1 (interaction and Ub-targeting) and the effector suggests that these separate domains are in a suitable orientation of the active or partially active /primed? NLR form, consistent with roles of SD and NB-LRR in NLR auto-inhibition and release.

Referee #2:

Nucleotide-binding leucine-rich repeat receptors (NLRs) represent the largest class of immune receptors in plants. They play a crucial role in the plant's defense system by detecting pathogen effectors that are delivered into the plant cell. Plants have

evolved a intricated regulatory mechanism to tightly control NLRs-mediated immunity in order to minimize fitness costs. Wang et al.'s study further demonstrates the significant role of ubiquitination in regulating the homeostasis of NLR proteins to ensure normal immune responses. They utilized the Sw-5b NLR as a model and identified an E3 ligase, SBP1, capable of ubiquitinating Sw-5b. Through genetic, biochemical, and molecular biology approaches, they confirmed that SBP1 regulates the turnover of both inactive and self-active forms of Sw-5b by targeting its N-terminal and NB-LRR domains. Additionally, they found that the tomato spotted wilt orthotospovirus (TSWV)-encoded NSm effector can counteract SBP1-mediated negative regulation of Sw-5b, thereby ensuring the proper activation of Sw-5b-mediated immunity. Overall, this study sheds light on the intricate regulation of NLR protein turnover during pathogen attack, adding to the understanding of the role of E3 ligases in modulating NLR function.

Major concerns :

1. In lines 138-139, besides the author's claim that SBP1 has a faster degradation rate for the autoactive form of Sw-5b and that NSm effector can alleviate SBP1-mediated degradation of Sw-5b, considering the need to promptly deactivate the immune system after pathogen clearance to prevent potential damage to the plant, I wonder whether SBP1 also plays a role in timely turn-off immunity during the late stages of immune activation? The authors could consider performing a time-course analysis of Sw-5b accumulation in wild-type and SBP1 knockout transgenic plants co-expressing Sw-5b and NSm. This would help to comprehensively analyze at which stage SBP1 acts on Sw-5b-mediated immunity.
2. How about the difference in the expression levels between YFP-Sw-5b and YFP-Sw-5bHeinz in wild-type and SBP1 knockout *N. benthamiana* leaves?
3. In Figure 1, are there any NSm mutants that do not interact with Sw-5b or are not recognized by Sw-5b, and how about their presence affect the stability of Sw-5b as well as its domains?
4. Was immunoprecipitation (IP) conducted before all immunoblot analyses of Sw-5b and SD protein? For example, are the transiently expressed proteins shown in Figure 1 subjected to IP before immunoblot analyses? Why is IP performed first?
5. In Figure 2, why do the accumulation levels of SD and NB-LRR proteins increase in the presence of SBP1RM relative to the control RFP in panels 2C and 2D, but the accumulation of Sw-5b does not vary significantly in the presence of SBP1RM and RFP in panel 2E? Furthermore, why does the protein accumulation of SBP1RM-RFP in the Input increase in the presence of NB-LRR and Sw-5b?
6. What about SBP1 expression during the activation of Sw-5b-mediated immunity? A time-course analysis could be performed to check the mRNA level of SBP1.
7. Wang et al. constructed several chimeric Sw-5b variants by aligning the amino acid sequences of Sw-5b and Sw-5bHeinz (Appendix Figure S7). How about the ubiquitination of these chimeric Sw-5b variants? Additionally, although the authors performed amino acid sequence alignment to identify possible regions responsible for the ubiquitination of Sw-5b by SBP1 (Appendix Figure S10), it remains unclear the key amino acids within Sw-5b that was ubiquitinated by SBP1. I think determination of these key amino acids (maybe lysine) is important to establish the genetic and biochemical link between SBP1 and Sw-5b.
8. Does the viral NSm interact with SBP1 or undergo ubiquitination by SBP1?
9. In line 317, the term "prevents" should be modified to reflect the results in Figure 5. Despite the extensive competitive co-IP experiments conducted by the authors to analyze the interference of the NSm effector with SBP1-Sw-5b interaction, to strengthen the conclusion, I suggest the authors perform competitive split-LUC experiments and *in vitro* competitive pull-down experiments with some combinations to make the conclusion more solid.
10. In Figure 3, are there any differences in the phenotype of necrosis induced by Sw-5b recognition of effector NSm in WT and *sbp1-1* or *sbp1-2* plants? Are there any differences in the severity or timing of necrosis? Similarly, does the ability of Sw-5bD857V to induce necrosis differ in WT and *sbp1-1* or *sbp1-2* plants?
11. In Figure 4, the specific location of the ubiquitination bands should be indicated on the blot panels. In panel 4D, the left and right panels are not consistent, especially in the rightmost lane of the right panel, where the intensity of the ubiquitination band in *sbp1-1* is stronger than in WT (INPUT, IB: YFP). Additionally, why does the expression level of YFP-Sw-5b increase in WT leaves upon expression of FLAG-Ub in the Input compared to when FLAG-Ub is not expressed?
12. In Figures 4-5, does the presence of viral NSm reduce the ubiquitination of Sw-5b by SBP1, either *in vivo* or *in vitro*?
13. Will tagging YFP at the N-terminus of Sw-5b affect its function? The authors sometimes use a Flag tag and sometimes a YFP tag, with the Flag tag added to the C-terminus of the SD domain (Figure 2C) and the YFP tag added to the N-terminus of the SD domain (Figure 3C). What is the rationale behind this? Could it affect the normal function of the SD domain?
14. In Figure 6, are there differences in the degradation rates of Sw-5b in WT and *sbp1-1* or *sbp1-2* tobacco plants? Are there any differences in the degradation rates of Sw-5bD857V and other chimeric Sw-5b variants in WT and *sbp1-1* or *sbp1-2* plants?
15. In Appendix Figure S3, besides the transgenic plants with knocked-out SBP1 (*sbp1-1* or *sbp1-2*), did the authors prepare transgenic plants overexpressing SBP1 and analyze their corresponding functions and phenotypes?
16. In Appendix Figure S7, does the severity of necrosis differ in panels H-J? If the accumulation of chimeric Sw-5b variants is higher, should the necrosis induced by the self-active mutant variants of Sw-5b (Sw-5bD857V/LRRM3-6 or Sw-5bD857V) be stronger relative to unmutated Sw-5bD857V? Additionally, evidence of mutations in ubiquitination sites (amino acids) on Sw-5b could further elucidate the impact of ubiquitination on Sw-5b-mediated immunity. Furthermore, in the immunoblot panel below the leaves, the differences in protein levels of Sw-5b and Sw-5bD857V fluctuates among the three blot panels. Moreover, in panels D-J, if both the SD and LRR domains indicated in F (ii and iii) were substituted simultaneously, what would be the stability and cell death phenotype of this chimeric variant?
17. In Appendix Figure S8, besides the phenotypes of various combinations shown in WT leaves, how about the necrosis phenotypes induced by various combinations in *sbp1-1* or *sbp1-2* leaves? Additionally, conductivity should be measured or

recorded to make the phenotype data more solid.

18. In line 349, does SBP1 interact with the LRR domain of Sw-5b and directly degrade the LRR domain? There seems only data about the NB-LRR domain without evidence for the LRR domain.

Minor concerns:

1. In Line 33, I suggest changing "self-active" to "autoactive".
2. In Lines 155 and 202, after the first appearance of "*Nicotiana benthamiana*", it can be abbreviated as "*N. benthamiana*" for brevity in the subsequent descriptions.
3. In Lines 173-174, should "Fig. 1D" be corrected to "Fig. 1C"?
4. In Lines 245 and 654, "qRT-PCR" should be written out in full when it first appears.
5. In Lines 355-356, I feel this sentence is somewhat redundant because the structure of the resistosome has already explained the transition of NLR from the resting state to the activation state. The conformational changes in NLRs during immune activation is common.
6. In Line 720, it should be "Student's *t*-test ($*P < 0.05$)." with a lowercase "t" and italicized "P".
7. In the legend to Fig. 1K, "21 dpi" should be corrected to "21 hpi".
8. In Figure 6F, I suggested changing "SDHeinz-CNLD857V" to "SDHeinz-CNLSw-5b/D857V" so as to correspond with what is shown in Figure 6E.

Referee #3:

In this report the authors show data that partly demonstrates that the plant NLR Sw-5b in the active state is turned over mediated by the E3 ligase SBP1 and the viral effector NSm blocks the interaction with the Ubiquitin E3 ligase to stabilise it and trigger immunity. The manuscript is written very clearly and the data is presented well. However there are clear shortcomings, of which I have listed a few below. I hope this will improve the manuscript.

1. Figure 1A. The stability assays need to be performed with cycloheximide treatment to ensure that it is crucially a post-translational effect on sw5b. this cycloheximide treatments should be included in all the stability assays in the manuscript including those in Figure 3.
2. It would be important to show that in MG132 treatment - that sw5b is tagged with ubiquitin. This is also the case for the SD and NB-LRR domains for sw5b. This data will substantiate their claim that the degradation is indeed through the ubiquitination pathway.
3. Figure 2, the authors should include the list of interactors that was identified in the Y2H screen and provide an explanation as to why they choose SBP1.
4. Figure 4, Ub assays the authors should include sw-5bHeinz as a control for specificity so that to substantiate their claim that SBP1 targets specifically sw-5b.
5. Figure 5 - to demonstrate that the Viral NSm effector interferes with SNBP1-sw5b interaction the authors should do competition assays with increasing concentration of NSM to show a dose dependent effect on the inhibitory effect on the interaction with the E3 ligase.

Nevertheless the manuscript might represent a very clear advance in the field if the biochemistry is undertaken in a more robust manner.

Response to Referee's Comments

We thank the referees for their time and constructive comments on our manuscript. Below is our point-by-point response to referees' comments.

Response to Referee #1:

Plant NLR immune receptors are usually triggered, directly or indirectly, by pathogen effectors and in the absence of pathogen attack various transcriptional and post-transcriptional mechanisms (including E3 Ub ligase turnover) have been characterized which keep NLRs at low levels and inactive to reduce fitness trade-offs. This study investigates NLR homeostatic processes regulating the function of a tomato CC-domain NLR receptor, Sw-5b, building on previous published insights by this group on Sw-5b functional activation by a viral effector. This is an interesting question because, although it is known that multi-domain NLR proteins can shift from auto-inhibited to primed (intermediate) and fully active (immune triggering) states, the processes underlying NLR state equilibrium are not fully resolved. The work presented here goes significantly beyond earlier studies and is important for the field. It demonstrates in a comprehensive and convincing way how the on/off state of the Sw-5b SD-CC-NB-LRR protein is controlled by a Ub E3 ligase (SBP1). Two aspects make the analysis interesting and novel: (i) characterizing the genetic and molecular basis of SBP1 differential control of Sw-5b turnover as an inhibited (inactive) form or as a conformationally activated form, and (ii) the mode of viral effector interference with SBP1-mediated degradation of Sw-5b through direct effector binding to Sw-5b domains to conformationally activate the NLR and compete with SBP1-NLR degradation - which leads to 'released' effector-triggered immunity. The data provide solid validation of previous NLR on-off equilibrium models, but also a detailed mode of action, by which different NLR states are maintained in healthy and pathogen-activated plants. The authors use complementary yeast, plant (Nb and tomato), in vitro assays and suitable controls (eg. an autoactive Ws-5b variant and SW-5b effector recognition competent and non-competent forms/chimeras) to test their model. The sum of the experiments is a compelling demonstration of processes underlying a CC-NLR state equilibrium and immune function. The Results are presented clearly and the Discussion is balanced and generally informative.

I don't have specific queries - more of a comment which authors might consider expanding on a bit more in Discussion. Although they don't have access to structurally resolved Sw-5b states, the fact that individual SD and NB-LRR domains are also targeted by SBP1 (interaction and Ub-targeting) and the effector suggests that these separate domains are in a suitable orientation of the active or partially active /primed? NLR form, consistent with roles of SD and NB-LRR in NLR auto-inhibition and release.

Response: We have expanded a bit more in Discussion including Sw-5b in active state undergoes a conformational change that allow the release of both SD and

NB-LRR, rendering them accessible to SBP1 for degradation (see lines 563-565); upregulation of E3 ligase SBP1 upon the activation of Sw-5b, which further enhances the protein turnover rate of the active form of Sw-5b (see lines 595-598); TSWV NSm but not non-elicitor TZSV NSm interferes with the ubiquitination of Sw-5b (see lines 534-539).

Response to Referee #2:

Nucleotide-binding leucine-rich repeat receptors (NLRs) represent the largest class of immune receptors in plants. They play a crucial role in the plant's defense system by detecting pathogen effectors that are delivered into the plant cell. Plants have evolved a intricated regulatory mechanism to tightly control NLRs-mediated immunity in order to minimize fitness costs. Wang et al.'s study further demonstrates the significant role of ubiquitination in regulating the homeostasis of NLR proteins to ensure normal immune responses. They utilized the Sw-5b NLR as a model and identified an E3 ligase, SBP1, capable of ubiquitinating Sw-5b. Through genetic, biochemical, and molecular biology approaches, they confirmed that SBP1 regulates the turnover of both inactive and self-active forms of Sw-5b by targeting its N-terminal and NB-LRR domains. Additionally, they found that the tomato spotted wilt orthotospovirus (TSWV)-encoded NSm effector can counteract SBP1-mediated negative regulation of Sw-5b, thereby ensuring the proper activation of Sw-5b-mediated immunity. Overall, this study sheds light on the intricate regulation of NLR protein turnover during pathogen attack, adding to the understanding of the role of E3 ligases in modulating NLR function.

Major concerns:

1. In lines 138-139, besides the author's claim that SBP1 has a faster degradation rate for the autoactive form of Sw-5b and that NSm effector can alleviate SBP1-mediated degradation of Sw-5b, considering the need to promptly deactivate the immune system after pathogen clearance to prevent potential damage to the plant, I wonder whether SBP1 also plays a role in timely turn-off immunity during the late stages of immune activation? The authors could consider performing a time-course analysis of Sw-5b accumulation in wild-type and SBP1 knockout transgenic plants co-expressing Sw-5b and NSm. This would help to comprehensively analyze at which stage SBP1 acts on Sw-5b-mediated immunity.

Response: It is a great IDEA that SBP1 plays a role in timely turn-off immunity during the late stages of immune activation. As suggested by this reviewer, we performed a time course analysis and found that SBP1 mRNA levels were upregulated during the activation of Sw-5b mediated immunity, suggesting that SBP1 likely involved in turn-off immunity (Figure EV 5D-E).

We have also performed a time-course analysis of Sw-5b accumulation in wild-type and SBP1 knockout *N. benthamiana* plants co-expressing Sw-5b and NSm. The

results showed that Sw-5b accumulation increased at 22 hpi and decreased at 24 and 26 hpi in wild-type *N. benthamiana* plants co-expressing Sw-5b and NSm. The onset of cell death induced by Sw-5b and NSm is about at 22-24 hpi. Sw-5b accumulation increased at 22 and 24 hpi and decreased at 26 hpi in SBP1 knockout *N. benthamiana* plants. These results indicate that SBP1 plays a role in timely turn-off immunity at late stage of immune activation (see the Supplemental file for review only; Supplemental Figure 1).

2. *How about the difference in the expression levels between YFP-Sw-5b and YFP-Sw-5bHeinz in wild-type and SBP1 knockout N. benthamiana leaves?*

Response: We have now determined the expression levels between YFP-Sw-5b and YFP-Sw-5bHeinz in wild-type and *SBP1* knockout *N. benthamiana* leaves. The results showed that the protein accumulation of YFP-Sw-5b was significantly elevated in *SBP1* knockout *N. benthamiana* leaves in comparison to the wild-type plant leaves. In contrast, the protein accumulation of YFP-Sw-5b^{Heinz} exhibited no significant difference between wild-type and *SBP1* knockout *N. benthamiana* leaves (see new Fig. EV1E).

3. *In Figure 1, are there any NSm mutants that do not interact with Sw-5b or are not recognized by Sw-5b, and how about their presence affect the stability of Sw-5b as well as its domains?*

Response: We have now used a tomato zonate spot orthotospovirus (TZSV) encoded NSm variant that is not recognized by Sw-5b (Zhu et al., Plant Cell, 2017). The results showed that the protein levels of Sw-5b as well as its SD and NB-LRR domains were not elevated by TZSV NSm (see new Fig. EV1A-C).

4. *Was immunoprecipitation (IP) conducted before all immunoblot analyses of Sw-5b and SD protein? For example, are the transiently expressed proteins shown in Figure 1 subjected to IP before immunoblot analyses? Why is IP performed first?*

Response: Because the Sw-5b NLR protein is expressed at extremely low levels, the protein of Sw-5b must be enriched by immunoprecipitation (IP) before it can be detected.

5. *In Figure 2, why do the accumulation levels of SD and NB-LRR proteins increase in the presence of SBP1RM relative to the control RFP in panels 2C and 2D, but the accumulation of Sw-5b does not vary significantly in the presence of SBP1RM and RFP in panel 2E? Furthermore, why does the protein accumulation of SBP1RM-RFP in the Input increase in the presence of NB-LRR and Sw-5b? (dominant negative effect)*

Response: That's probably because the expression levels of SD and NB-LRR proteins is higher than the full-length Sw-5b, which is really low. Although the amount of the full-length Sw-5b is increased in the presence of SBP1RM, the increased amount of the full-length Sw-5b seems not as pronounced as its SD and NB-LRR domains.

The addition of SBP1RM-RFP resulted in an increase in the accumulation of NB-LRR and Sw-5b. The SBP1RM mutant no longer contain E3 enzyme activity and the effect by SBP1RM is probably because this mutant has a dominant negative effect. The SBP1RM mutant may inhibit endogenous E3 enzyme activity of SBP1, which leads to an increase in the accumulation of NB-LRR and Sw-5b.

6. *What about SBP1 expression during the activation of Sw-5b-mediated immunity? A time-course analysis could be performed to check the mRNA level of SBP1.*

Response: Thank you very much for your insightful thoughts. To examine the *SBP1* expression during the activation of Sw-5b-mediated immunity, we performed a time-course analysis of *SBP1* mRNA levels in WT *N. benthamiana* plants expressing Sw-5b or Sw-5b^{D857V}. Meanwhile we analyzed *SBP1* mRNA levels in WT plants co-expressing Sw-5b and TSWV NSm and used plants co-expressing Sw-5b and nonelicitor TZSV NSm as a control. The results showed that the mRNA expression levels of *SBP1* were upregulated during the activation of Sw-5b-mediated immunity (see new Fig. EV5D, E).

7. *Wang et al. constructed several chimeric Sw-5b variants by aligning the amino acid sequences of Sw-5b and Sw-5bHeinz (Appendix Figure S7). How about the ubiquitination of these chimeric Sw-5b variants? Additionally, although the authors performed amino acid sequence alignment to identify possible regions responsible for the ubiquitination of Sw-5b by SBP1 (Appendix Figure S10), it remains unclear the key amino acids within Sw-5b that was ubiquitinated by SBP1. I think determination of these key amino acids (maybe lysine) is important to establish the genetic and biochemical link between SBP1 and Sw-5b.*

Response: We have now examined the ubiquitination of Sw-5b*Heinz* and the chimeric Sw-5b variants. The results showed that the ubiquitination of YFP-Sw-5b and YFP-Sw-5b^{LRRM3-6} was higher than that of YFP-SD^{Heinz}-(CC-NB-LRR)^{Sw-5b}. However, the ubiquitination of these chimeric Sw-5b variants was significantly reduced in the *SBP1* knockout mutant (see new Fig. EV4G). Moreover, the ubiquitination of YFP-Sw-5b^{D857V} was found to be higher than that of YFP-SD^{Heinz}-CNL^{Sw-5b/D857V} and YFP-Sw-5b^{D857V/LRRM3-6}. Similarly, the ubiquitination of these chimeric Sw-5b variants was significantly reduced in *SBP1* knockout plants (see new Fig. EV4G).

To determine the amino acids within Sw-5b that was ubiquitinated by SBP1, Sw-5b SD and NB-LRR domains ubiquitinated by SBP1 were analyzed by Mass spectrometry. We have determined K79, K81 and K153 in the SD domain were ubiquitinated using Mass spectrometry. The immunoblot results showed that the protein levels of YFP-SD^{K79/81/153R} were significantly increased in WT *N. benthamiana* leaves compared to WT YFP-SD, whereas these differences were not detectable in *SBP*-KO plants (Fig. EV2B). Attempts to identify ubiquitinated amino acids in NB-LRR by Mass spectrometry failed. We have previously identified the small region in LRR encompassing polymorphic sites 3-6 that are required for the

Nsm recognition (Zhu et al., 2017). To investigate whether SBP1 directly regulates the small region encompassing the Nsm recognition sites of LRR, we fused the 945-1055 amino acid of Sw-5b and Sw-5b^{Heinz} with YFP at the C-terminus (Fig. EV2C, D) and analyzed their protein levels in both WT and *sbp1-1* plants. As shown in Figure EV2D, the protein accumulation of Sw-5b⁹⁴⁵⁻¹⁰⁵⁵-YFP was lower than Sw-5b^{Heinz945-1055}-YFP in WT *N. benthamiana* plants, however their protein levels were similar in *sbp1-1* plants. There are nine lysines in Sw-5b LRR3-6 and scanning of these amino acids by arginine replacement showed that amino acid residues K1022 and K1027 are important for the ubiquitination of LRR3-6 by SBP1 (see new Figure EV2E).

8. Does the viral Nsm interact with SBP1 or undergo ubiquitination by SBP1?

Response: We didn't detect the interaction between Nsm and SBP1 in yeast two hybrid (Y2H) assays. We also detected the protein accumulation of Nsm in wild-type (WT) and *SBP1* knockout *N. benthamiana* plants. The results showed that there is no discernable difference on the Nsm protein levels in WT and *SBP1* knockout plants (see supplemental file for review only; Figure 2).

We also found that the addition of TSWV Nsm significantly diminished the ubiquitination of Sw-5b, however the addition of TZSV Nsm did not have such effect (see new Fig. 5C, D). Furthermore, the addition of TSWV Nsm neither influenced the overall ubiquitination of plant endogenous proteins *in vivo* nor the ubiquitination of SBP1-HA-His itself *in vitro* (see new Fig. 5C, D). These data suggest that TSWV Nsm interferes with the ubiquitination of Sw-5b by SBP1.

9. In line 317, the term "prevents" should be modified to reflect the results in Figure 5. Despite the extensive competitive co-IP experiments conducted by the authors to analyze the interference of the Nsm effector with SBP1-Sw-5b interaction, to strengthen the conclusion, I suggest the authors perform competitive split-LUC experiments and *in vitro* competitive pull-down experiments with some combinations to make the conclusion more solid.

Response: We have modified the text accordingly (see line 350).

We have also performed the competitive split-LUC experiments and competitive GST pull-down assays to investigate the impact of the Nsm effector on SBP1-Sw-5b SD/NB-LRR interactions. As illustrated in new Figure EV3C-F, when the OD₆₀₀ of *Agrobacterium* carrying TSWV Nsm was elevated from 0 to 0.5, the interactions between SBP1 and Sw-5b SD or between SBP1 and NB-LRR were markedly inhibited. This inhibition effect was not observed when the concentration of *Agrobacterium* carrying TZSV Nsm was increased from 0 to 0.5 of OD₆₀₀. The competitive GST pull-down assay results also showed that GST-Sw-5b exhibited a reduced capacity to bind SBP1 in the presence of TSWV Nsm, but not TZSV Nsm (see new Fig. 5B).

10. In Figure 3, are there any differences in the phenotype of necrosis induced by Sw-5b recognition of effector NSm in WT and *sbp1-1* or *sbp1-2* plants? Are there any differences in the severity or timing of necrosis? Similarly, does the ability of Sw-5b^{D857V} to induce necrosis differ in WT and *sbp1-1* or *sbp1-2* plants?

Response: We didn't observe the severity of the cell death induced by Sw-5b and NSm between WT and *sbp1-1* plants. The time-course (24, 30, 36, 42, 48 h) conductivity analysis showed that there was no discernible difference in conductivity induced by Sw-5b and NSm in WT and *sbp1-1* plants from 24 to 48 hpi (Fig. EV5F right panel). However, the conductivity in *sbp1-1* mutant plants treated with Sw-5b^{D857V} was higher than that in WT plants at 24-30 hpi and that there was no significant difference in conductivity between WT and *sbp1-1* plants at 36-48 hpi (Fig. EV5F left panel).

11. In Figure 4, the specific location of the ubiquitination bands should be indicated on the blot panels. In panel 4D, the left and right panels are not consistent, especially in the rightmost lane of the right panel, where the intensity of the ubiquitination band in *sbp1-1* is stronger than in WT (INPUT, IB: YFP). Additionally, why does the expression level of YFP-Sw-5b increase in WT leaves upon expression of FLAG-Ub in the Input compared to when FLAG-Ub is not expressed?

Response: Thank you for your helpful comments. We have indicated the ubiquitination bands on the blot panels (see new Figure 4 and Figure 5). In panel 4D, the left panel is the amount of ubiquitinated Sw-5b, while the right panel is the amount of input Sw-5b. We couldn't differentiate the ubiquitinated and non-ubiquitinated Sw-5b in the right panel. However, when Sw-5b was pulled down with FLAG-Ub, the amount of ubiquitinated Sw-5b was significantly reduced in *SBP1* knockout mutant.

It is intriguing that the expression level of YFP-Sw-5b was increased in WT leaves upon expression of FLAG-Ub in the Input compared to when FLAG-Ub is not expressed. One interpret is that overexpression of Ub with FLAG may interfere with the endogenous ubiquitination and degradation of YFP-Sw-5b level. These experiments were done in the presence of MG132. When FLAG-Ub is expressed, these Ub with FLAG tag were added to Sw-5b and their degradation was blocked by MG132.

12. In Figures 4-5, does the presence of viral NSm reduce the ubiquitination of Sw-5b by *SBP1*, either *in vivo* or *in vitro*?

Response: We have examined whether viral NSm can interfere with the ubiquitination of Sw-5b by *SBP1* *in vivo* and *in vitro*. The results showed that the addition of TSWV NSm significantly diminished the ubiquitination of Sw-5b, however the addition of TZSV NSm did not have such effect (see new Fig. 5C, D). Furthermore, the addition of TSWV NSm neither influenced the overall ubiquitination of plant endogenous proteins *in vivo* nor the ubiquitination of *SBP1*-HA-His itself *in vitro* (see new Fig. 5C, D).

13. Will tagging YFP at the N-terminus of Sw-5b affect its function? The authors sometimes use a Flag tag and sometimes a YFP tag, with the Flag tag added to the C-terminus of the SD domain (Figure 2C) and the YFP tag added to the N-terminus of the SD domain (Figure 3C). What is the rationale behind this? Could it affect the normal function of the SD domain?

Response: Tagging YFP at the N-terminus of Sw-5b did not affect its function. We have different version of Tag added to the SD domain. They didn't affect the normal function of the SD domain.

14. In Figure 6, are there differences in the degradation rates of Sw-5b in WT and *sbp1-1* or *sbp1-2* tobacco plants? Are there any differences in the degradation rates of Sw-5bD857V and other chimeric Sw-5b variants in WT and *sbp1-1* or *sbp1-2* plants?

Response: We have examined the differences in the degradation rates of Sw-5b in WT and *sbp1-1* plants. The results showed that the degradation of Sw-5b in WT plant is faster than that in *sbp1* plants. The protein turnover of the chimeric SD^{Heinz}-CNL^{Sw-5b/D857V} and Sw-5b^{D857V/LRRM3-6} mutants was slower than that of the Sw-5b^{D857V} in the WT *N. benthamiana* leaves. However, the protein turnover rate of these chimeric mutants was also significantly reduced in the *sbp1-1* mutant (Fig. EV5A-C).

15. In Appendix Figure S3, besides the transgenic plants with knocked-out SBP1 (*sbp1-1* or *sbp1-2*), did the authors prepare transgenic plants overexpressing SBP1 and analyze their corresponding functions and phenotypes?

Response: We didn't prepare the transgenic plants overexpressing SBP1, but we have analyzed the HR cell death and the ion leakage phenotypes of Sw-5b^{D857V}, and Sw-5bD857V+NSm in the absence or the presence of SBP1 in WT *N. benthamiana* leaves. The results showed that coexpression of SBP1 with Sw-5b^{D857V} significantly reduced the cell death induced by Sw-5b^{D857V}. Coexpression of SBP1 also reduced the protein accumulation of Sw-5b^{D857V}. However, NSm can stabilize Sw-5b^{D857V} from the degradation by SBP1 and co-expression of Sw-5b^{D857V}, SBP1 and NSm induce severe cell death (Fig. EV5G, H).

16. In Appendix Figure S7, does the severity of necrosis differ in panels H-J? If the accumulation of chimeric Sw-5b variants is higher, should the necrosis induced by the self-active mutant variants of Sw-5b (Sw-5bD857V/LRRM3-6 or Sw-5bD857V) be stronger relative to unmutated Sw-5bD857V? Additionally, evidence of mutations in ubiquitination sites (amino acids) on Sw-5b could further elucidate the impact of ubiquitination on Sw-5b-mediated immunity. Furthermore, in the immunoblot panel below the leaves, the differences in protein levels of Sw-5b and Sw-5bD857V fluctuates among the three blot panels. Moreover, in panels D-J, if both the SD and LRR domains indicated in F (ii and iii) were substituted simultaneously, what would be the stability and cell death phenotype of this chimeric variant?

Response: We didn't observe that the necrosis induced by the self-active mutant

variants of Sw-5b (Sw-5bD857V/LRRM3-6 or Sw-5bD857V) is stronger than Sw-5bD857V (see new Appendix Figure S7E). This is probably because the conformation of chimeric Sw-5b mutant itself also affect their ability to induce cell death. We have also substituted both the SD and LRR domains simultaneously, although the stability of this mutant was increased, we didn't observe the severe necrosis either (new Appendix Figure S7D). The 3D structural conformation of chimeric Sw-5b-D857V mutant may also affect its ability to induce cell death. We have included a discussion for these mutants in the revised manuscript (see lines 576-580).

17. In Appendix Figure S8, besides the phenotypes of various combinations shown in WT leaves, how about the necrosis phenotypes induced by various combinations in sbp1-1 or sbp1-2 leaves? Additionally, conductivity should be measured or recorded to make the phenotype data more solid.

Response: We have measured the conductivity for the various combinations (see new Appendix Figure S8E, F).

18. In line 349, does SBP1 interact with the LRR domain of Sw-5b and directly degrade the LRR domain? There seems only data about the NB-LRR domain without evidence for the LRR domain.

Response: LRR alone is difficult to be expressed. We haven't been able to check the degradation of the LRR domain.

We found that LRR-3-6 NSm recognition sites fused with YFP was indeed regulated by SBP1. Protein accumulation of Sw-5b⁹⁴⁵⁻¹⁰⁵⁵-YFP was significantly higher in *SBP1* knockout plant leaves than that in WT leaves. However, the protein accumulation of Heinz LRR⁹⁴⁵⁻¹⁰⁵⁴-YFP had no significant difference between WT and *SBP1* knockout plant leaves (see new Figure EV2C, D). We have also determined that amino acid residues K1022 and K1027 are important for the ubiquitination of LRR3-6 by SBP1 (see new Figure EV2E).

Minor concerns:

- 1. In Line 33, I suggest changing "self-active" to "autoactive".*
- 2. In Lines 155 and 202, after the first appearance of "Nicotiana benthamiana", it can be abbreviated as "N. benthamiana" for brevity in the subsequent descriptions.*
- 3. In Lines 173-174, should "Fig. 1D" be corrected to "Fig. 1C"?*
- 4. In Lines 245 and 654, "qRT-PCR" should be written out in full when it first appears.*
- 5. In Lines 355-356, I feel this sentence is somewhat redundant because the structure of the resistosome has already explained the transition of NLR from the resting state to the activation state. The conformational changes in NLRs during immune activation is common.*
- 6. In Line 720, it should be "Student's t-test (*P<0.05)." with a lowercase "t" and italicized "P".*
- 7. In the legend to Fig. 1K, "21 dpi" should be corrected to "21 hpi".*

8. In Figure 6F, I suggested changing "SDHeinz-CNLD857V" to "SDHeinz-CNLSw-5b/D857V" so as to correspond with what is shown in Figure 6E.

Response: Thank you very much for pointing out these errors. We have corrected them all accordingly.

Response to Referee #3:

In this report the authors show data that partly demonstrates that the plant NLR Sw-5b in the active state is turned over mediated by the E3 ligase SBP1 and the viral effector NSm blocks the interaction with the Ubiquitin E3 ligase to stabilise it and trigger immunity. The manuscript is written very clearly and the data is presented well. However there are clear shortcomings, of which I have listed a few below. I hope this will improve the manuscript.

1. Figure 1A. The stability assays need to be performed with cycloheximide treatment to ensure that it is crucially a post-translational effect on sw-5b. this cycloheximide treatments should be included in all the stability assays in the manuscript including those in Figure 3.

Response: Thanks for this reviewer's helpful suggestions. For all stability assays, we have examined their transcription levels and found no difference in mRNA levels. To further strengthen the conclusion, we have performed the stability assays with cycloheximide treatment for Figure 1A, 1F, 1G, Figure 3A-D and other necessary Figures (see new Appendix Figure S1A-C, Expanded View Figure 1A-G, and Expanded View Figure 4D, F).

2. It would be important to show that in MG132 treatment - that sw5b is tagged with ubiquitin. This is also the case for the SD and NB-LRR domains for sw5b. This data will substantiate their claim that the degradation is indeed through the ubiquitination pathway.

Response: The *in vivo* ubiquitin assay in Figure 4D was performed in the presence of MG132. This is indicated in the Methods. We have now indicated the MG132 treatment in the Figure 4D. As suggested by Reviewer #3, we have further performed ubiquitination assays for the SD and NB-LRR domains for Sw-5b with the MG132 treatment in WT and *sbp1-1* mutant *N. benthamiana* leaves. As shown in new Figure 4E and 4F, both SD and NB-LRR domains were ubiquitinated in WT plants, while these ubiquitinations were significantly reduced in the *sbp1-1* mutants.

3. Figure 2, the authors should include the list of interactors that was identified in the Y2H screen and provide an explanation as to why they choose SBP1.

Response: We have included the list of interactors (see new Appendix Figure S2A). As reason for competition, the exact name for the interactors were not shown. Among these interactors, SBP1 is the only E3 Ub ligase that we selected in Y2H screen.

4. *Figure 4, Ub assays the authors should include sw-5bHeinz as a control for specificity so that to substantiate their claim that SBP1 targets specifically sw-5b.*

Response: We have now used Sw-5b^{Heinz} as a control for specificity. The results showed that there was no difference in the ubiquitination of Sw-5b^{Heinz} between WT and *SBP1* knockout *N. benthamiana* leaves (see new Fig. EV2A).

5. *Figure 5 - to demonstrate that the Viral NSm effector interferes with SNBP1-sw5b interaction the authors should do competition assays with increasing concentration of NSM to show a dose dependent effect on the inhibitory effect on the interaction with the E3 ligase.*

Response: We have performed the competitive split-LUC experiments and competitive GST pull-down assays with increasing concentration of NSm. The results indeed show a dose dependent effect of TSWV NSm on the interaction between the SBP1 and Sw-5b (see new Fig. 5B; Figure EV3C-F).

Dear Xiaorong,

We have now received re-review reports from two referees, which I have included below. As you will see, you have addressed their concerns satisfactorily; however, I would like you to consider addressing their remaining points in the discussion section. Before I can finally accept the manuscript, there are some remaining editorial points which need to be addressed. In this regard would you please:

- acknowledge the following funding in our online submission system: 2022YFD1401200, the Funds from the Independent Innovation of Agricultural Science and Technology of Jiangsu Province [Grant No. CX (22)2039], the Jiangsu Key Technology R & D Program and International Science and Technology Cooperation Project (BZ2023030), the Guidance Foundation of the Sanya Institute of Nanjing Agricultural University (NAUSY-MS19), and the Key Science and Technology Program of Hainan Province (ZDKJ2021007),
- include a "Disclosure and competing interests statement",
- remove the AC/CrediT section from the text,
- rename "Table S1" as "Appendix Table S1", and "Supplementary Table 1" as "Appendix Table S1",
- include page numbers for each Appendix item on the title page, and remove Appendix figure legends from the manuscript file,
- complete the uploaded source data checklist; group source data into one folder per figure (within this folder there should be a single file for each panel); source data for EV and Appendix figures can be grouped into one zipped folder, refer to figures 1b-g, and EV 3d-e in a sequential manner, and
- define the annotated p values *****/**/***/a/b in the legends of figures EV 3d, f; EV 5d-f, h; also provide the exact p-value for the same, as appropriate.

We include a synopsis of the paper (see <http://emboj.embopress.org/>). Please provide me with a general summary image, two sentence statement and 3-5 bullet points that capture the key findings of the paper.

I am looking forward to receiving your revised manuscript.

EMBO Press is an editorially independent publishing platform for the development of EMBO scientific publications.

Best wishes,

William

We realize that it is difficult to revise to a specific deadline. In the interest of protecting the conceptual advance provided by the work, we recommend a revision within 3 months (11th Sep 2024). Please discuss the revision progress ahead of this time with the editor if you require more time to complete the revisions. Use the link below to submit your revision:

Referee #1:

In this revised manuscript the authors have responded quite comprehensively to Rev. #2 and #3 critical comments. I think the chief claims are now more strongly supported with detailed analysis of Sw-5b protein turnover states and cell death timing in the 'auto-active' and effector-triggered NLR responses. Overall, this is a well-executed study with informative controls which (as I mentioned for the initial submission) I think will be of broad interest for the field. The revision is generally well written and clear but I suggest authors:

1. denote the Sw5b-eliciting and non-eliciting forms of TZSV NSm differently throughout text so it's clear what they're referring to.
2. Change the title to reflect the key finding more accurately eg. 'Competition between a host E3 ligase and viral pathogen effector in controlling turnover and immune activity of plant NLR receptor Sw-5b' ... or similar.

Referee #2:

Homeostasis of NLR proteins is crucial for maintaining the balance between growth and plant defense, and ubiquitination plays an important role in regulating the stability of NLR proteins. For example, several NLR proteins have been reported to be finely regulated by ubiquitination, such as SNC1 (Cheng et al., 2011, PNAS; Dong et al., 2018, Nature Plants; Wu et al., 2020, EMBO Journal) and MLA (Wang et al., 2016, Plant Physiology). This study identified the E3 ligase SBP1 that mediates the ubiquitination of a plant NLR, Sw-5b, and demonstrated that SBP1 can regulate the turnover of Sw-5b, while NSm can disrupt their interaction to ensure robust immunity. Overall, this study extends the role of ubiquitination in regulating the homeostasis of NLR proteins and presents some interesting points. In the revised manuscript, Wang et al. performed a series of additional experiments to address my concerns. Overall, the manuscript quality has improved after revision. Although there are still some minor points in the article that are difficult to understand or explain, this work still provides some new insights for understanding the intricate regulation of NLR protein homeostasis.

I just have one more question that needs the author to address. Generally speaking, from an evolutionary perspective, pathogens always utilize virulence factors to counteract the plant defense system so as to achieve their infections. However, in this study, it seems that the virus-encoded NSm effector help the plants establish better defense by stabilizing the Sw-5b NLR protein, which would be detrimental to its own infection. I wonder how to understand this seemingly contradictory result from an evolutionary perspective. Why would the virus evolve NSm to stabilize Sw-5b, thereby being detrimental to itself ?

Response to Referee's Comments

We thank again the referees for their time and constructive comments on our manuscript. Below is our point-by-point response to referees' comments.

Response to Referee #1:

In this revised manuscript the authors have responded quite comprehensively to Rev. #2 and #3 critical comments. I think the chief claims are now more strongly supported with detailed analysis of Sw-5b protein turnover states and cell death timing in the 'auto-active' and effector-triggered NLR responses. Overall, this is a well-executed study with informative controls which (as I mentioned for the initial submission) I think will be of broad interest for the field. The revision is generally well written and clear but I suggest authors:

1. denote the Sw5b-eliciting and non-eliciting forms of TZSV NSm differently throughout text so it's clear what they're referring to.

Response: We have denoted the Sw5b-eliciting and non-eliciting forms of NSm accordingly (see highlights in the revised manuscript).

2. Change the title to reflect the key finding more accurately eg. 'Competition between a host E3 ligase and viral pathogen effector in controlling turnover and immune activity of plant NLR receptor Sw-5b' ... or similar.

Response: Thank you very much for your helpful suggestions. We have changed the title to “*Effector reverses E3-mediated turnover of plant NLR receptor to trigger robust immunity*”.

Response to Referee #2:

Homeostasis of NLR proteins is crucial for maintaining the balance between growth and plant defense, and ubiquitination plays an important role in regulating the stability of NLR proteins. For example, several NLR proteins have been reported to be finely regulated by ubiquitination, such as SNCI (Cheng et al., 2011, PNAS; Dong et al., 2018, Nature Plants; Wu et al., 2020, EMBO Journal) and MLA (Wang et al., 2016, Plant Physiology). This study identified the E3 ligase SBP1 that mediates the ubiquitination of a plant NLR, Sw-5b, and demonstrated that SBP1 can regulate the turnover of Sw-5b, while NSm can disrupt their interaction to ensure robust immunity. Overall, this study extends the role of ubiquitination in regulating the homeostasis of NLR proteins and presents some interesting points. In the revised manuscript, Wang et al. performed a series of additional experiments to address my concerns. Overall, the manuscript quality has improved after revision. Although there are still some minor points in the article that are difficult to understand or explain, this work still provides some new insights for understanding the intricate regulation of NLR protein

homeostasis.

I just have one more question that needs the author to address. Generally speaking, from an evolutionary perspective, pathogens always utilize virulence factors to counteract the plant defense system so as to achieve their infections. However, in this study, it seems that the virus-encoded NSm effector help the plants establish better defense by stabilizing the Sw-5b NLR protein, which would be detrimental to its own infection. I wonder how to understand this seemingly contradictory result from an evolutionary perspective. Why would the virus evolve NSm to stabilize Sw-5b, thereby being detrimental to itself ?

Response: Thank you for your insightful thoughts. NSm converted Sw-5b to trigger robust immunity maybe indeed not appropriate. As ETI is effector triggered immunity, we have changed the title to “***Effector reverses E3-mediated turnover of plant NLR receptor to trigger robust immunity***” to fit it better with ETI.

Dear Xiaorong,

I am pleased to inform you that your manuscript has been accepted for publication in the EMBO Journal.

Congratulations to you and you co-authors on this elegant study!

Yours sincerely,

William

William Teale, PhD
Editor
The EMBO Journal
w.teale@embojournal.org
